# Reward-Instruct: A Reward-Centric Approach to Fast Photo-Realistic Image Generation

**Yihong Luo**[1]*, **Tianyang Hu**[2]*, **Weijian Luo**[3], **Kenji Kawaguchi**[4], **Jing Tang**[5,1]†
[1]HKUST [2]CUHK(SZ) [3]Xiaohongshu Inc [4]NUS [5]HKUST(GZ)

## Abstract

This paper addresses the challenge of achieving high-quality and fast image generation that aligns with complex human preferences. While recent advancements in diffusion models and distillation have enabled rapid generation, the effective integration of reward feedback for improved abilities like controllability and preference alignment remains a key open problem. Existing reward-guided post-training approaches targeting accelerated few-step generation often deem diffusion distillation losses indispensable. However, in this paper, we identify an interesting yet fundamental paradigm shift: as conditions become more specific, well-designed reward functions emerge as the primary driving force in training strong, few-step image generative models. Motivated by this insight, we introduce **Reward-Instruct**, a novel and surprisingly simple reward-centric approach for converting pre-trained base diffusion models into reward-enhanced few-step generators. Unlike existing methods, Reward-Instruct does not rely on expensive yet tricky diffusion distillation losses. Instead, it iteratively updates the few-step generator's parameters by directly sampling from a reward-tilted parameter distribution. Such a training approach entirely bypasses the need for expensive diffusion distillation losses, making it favorable to scale in high image resolutions. Despite its simplicity, Reward-Instruct yields surprisingly strong performance. Our extensive experiments on text-to-image generation have demonstrated that Reward-Instruct achieves state-of-the-art results in visual quality and quantitative metrics compared to distillation-reliant methods, while also exhibiting greater robustness to the choice of reward function.

## 1 Introduction

High-quality, controllable, and fast image generation stands as a paramount goal in the field of Artificial Intelligence Generated Content (AIGC). Recent advancements in diffusion models [57, 18, 53] and, particularly, diffusion distillation [56, 28] have yielded impressive few-step image generators capable of rapid synthesis of photo-and-movie-realistic images and videos [47, 48, 68, 30, 32]. While these advancements have significantly improved generation speed and visual fidelity, how to achieve improved **controllability** and **alignment with complex human preferences** remains challenging.

Inspired by the success of reinforcement learning (RL) and more general reward-driven methodologies in large-language models [73, 36, 1], the image generation community has made considerable strides in developing effective *reward functions* for images. These rewards, broadly include any discriminative model capable of evaluating image quality or adherence to specific criteria, offer a promising avenue for guiding generative models towards desired attributes, such as human preferences, instruction following, as well as safety. Yet, the optimal strategies for effectively integrating and leveraging these reward signals in image generation workflows are still actively being investigated.

---

*Equal contribution.
†Corresponding author: Jing Tang.

39th Conference on Neural Information Processing Systems (NeurIPS 2025).

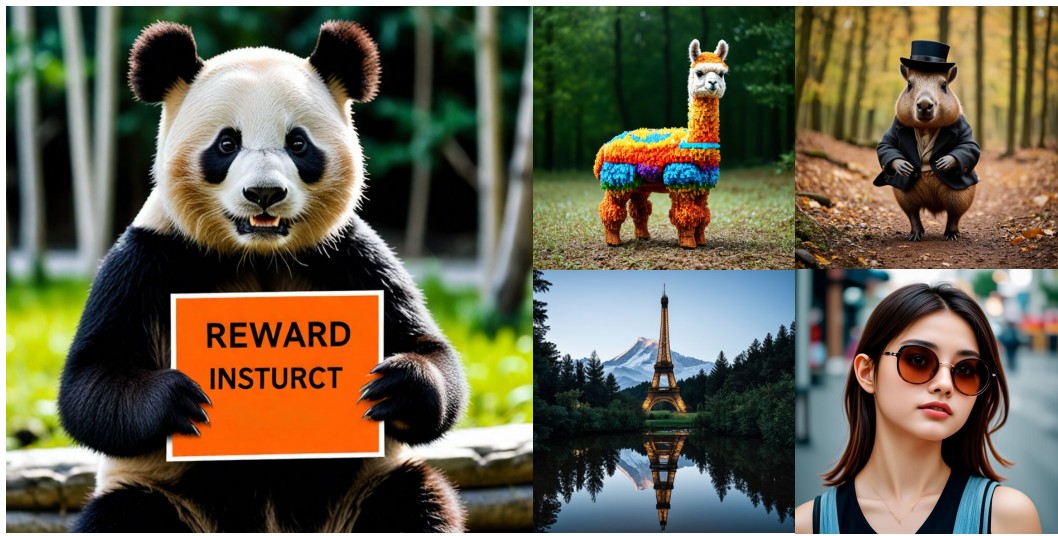

Figure 1: 4-step samples at 1024 resolution generated by **Reward-Instruct**. The **Reward-Instruct** here is trained from `SD3-medium` purely by reward maximization.

Current standard practices for incorporating reward signals into image generation pipelines predominantly occur as a post-training stage. These methods generally fall into two main categories: *(1).* integrating reward optimization directly into a pre-trained diffusion model, often followed by a distillation process for efficient sampling, as exemplified by SDE control and other reward-based post-training techniques [13, 8]; *(2).* Developing methods that post-train some already distilled, fast sampling models by applying reward fine-tuning methods together with expensive diffusion distillation losses. Typical works on this line are Diff-Instruct++ (DI++) [27]. However, both lines of approaches are considered computationally demanding and not inherently designed for directly converting pre-trained base diffusion models into few-step, reward-enhanced generators. Furthermore, despite the seemingly good metrics on benchmarks, *close examination of state-of-the-art reward-enhanced few-step generators reveals evidence of reward hacking* — certain artifacts or repeated objects in the background, as illustrated in Fig. 2.

Achieving reward-enhanced fast image generation hinges on two key driving forces: the knowledge embedded within the base pre-trained generative model and the knowledge derived from reward signals. While language modeling has demonstrated the primacy of the latter, reward signals, such reward-centric approaches remain underexplored in image modeling, as diffusion distillation losses are often considered indispensable in current works. With these observations, we are strongly motivated by an important scientific problem:

- ***Can we develop a reward-centric training approach, that can result in fast generation speed without the need of tricky yet expensive diffusion distillation losses?***

In this paper, we give a positive answer to this question and introduce a reward-centric method for training few-step generative models. Our journey starts with an analysis in Section 3, which shows that the post-training of DI++ is largely driven by reward signals, rather than the diffusion distillation objective for preserving knowledge of the image distribution from pretrained base models. This points towards a *phase transition* to modern conditional generation tasks — the increasing specificity and diversity of desired conditions or reward signals are causing a fundamental shift in the generation process, moving away from primarily modeling the conditional distribution and towards regularized reward maximization.

Taking this perspective to its logical conclusion, we propose a novel reward-centric approach termed **Reward-Instruct (RI)**, which is capable of efficiently training few-step generative models using reward. Fundamentally differs from existing approaches [27, 23], our methods directly operate on the base diffusion model and performs a direct reward maximization with simple yet effective regularization without the explicit requirement of a separate distillation loss or training images. Specifically, we start from a few-step sampler from pretrained diffusion models with enhanced stochasticity (via random $\eta$ sampling in Section 3.2.1) at each step. Our subsequent training process can be conceptually understood as directly sampling from a reward-tilted distribution within the

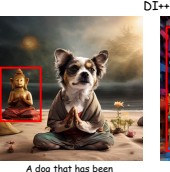
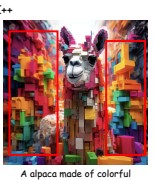
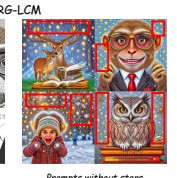
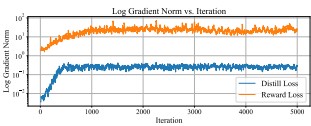

DI++           RG-LCM

A dog that has been meditating all the time    A alpaca made of colorful building blocks, cyberpunk    A photo of a monkey wearing glasses in a suit    Prompts without stars

Figure 2: Samples are taken from the corresponding papers of DI++ and RG-LCM. It can be observed that certain artifacts exist in samples, e.g., repeated text/objects in the background. We hypothesize this comes from *reward hacking*.

Figure 3: Log Gradient norm curve of DI++ through training process. We use the best configuration reported in DI++ [27].

generator's parameter space. Such a simple reward-centric design significantly improves the training efficiency, training stability, resulting in state-of-the-art results in few-step text-to-image generation with high visual quality (Fig. 1). Particularly, our method outperforms previous methods that combine diffusion distillation and reward learning regarding visual quality and machine metrics (Fig. 5 and Table 1), while being more robust to the reward choice (Fig. 6).

## 2    Preliminary

**Diffusion Models.** Diffusion models (DMs) [53, 18, 57] define a forward diffusion process that $x_t = \alpha_t x + \sigma_t \epsilon$, where $x$ is sampled from the data distribution, and $\sigma_t$ specifies the noise schedule. By training a denoising function to predict the added noise, the model implicitly learns the score of the data distribution, enabling the generation of new samples by simulating a reverse stochastic differential equation or ordinary differential equation [57, 54, 25, 70, 65]. Conditional generation is often achieved through techniques like classifier-free guidance (CFG), which modulates the denoising process based on the desired conditions [57, 54, 25, 70, 65, 31]. As illustrated in [27], CFG can be seen as an implicit reward on condition alignment.

**Preference Alignment with Diffusion Distillation.** Currently several methods [27, 29, 23] have been proposed for developing preference align few-step text-to-image models. Their approach can be summarized as a combination of distillation loss and reward loss: $\min_\theta L(\theta) = L_{\text{distill}}(x_\theta) - R(x_\theta, c)$, where $x_\theta$ denotes the model samples and $R(x_\theta, c)$ denotes the reward measure the alignment between $x_\theta$ and condition $c$. The distillation loss can be consistency distillation [23] or reverse-KL divergence [27]. The previous method either requires real data for training [23], or requires training an extra online score model [27]. These components increase the complexity of the post-training and serve solely as an overly complicated regularization.

## 3    Reward-Instruct: A Reward-Centric Approach to Image Generation

### 3.1    Discriminative vs Generative: The Phase Transition

The concerning phenomenon of *reward hacking*, as visually evidenced in Fig. 2, strongly suggests a potential imbalance where the generation process becomes excessively driven by reward models, potentially at the expense of image quality and generalization. To further verify this, our examination of the gradient norms of the reward loss and the distillation loss in our re-implementation of DI++ [27], shown in Figure 3, clearly indicates that updates in DI++ are overwhelmingly dominated by the reward term, relegating the diffusion distillation objective to a secondary role.

Extending this observation, we observe similar phenomena when considering a generalized concept of rewards that encompasses any discriminative model capable of judging the goodness of generated samples. In modern text-to-image generation models, various guidance modules for aligning conditions [19, 2, 33] are disproportionately amplified compared with the seemingly more important diffusion generation component. For instance, large CFG coefficients are indispensable (7.5 by default in Stable Diffusion [46] and 100 in DreamFusion [42]). When dealing with strong conditions, such as generating an image of "a red cube on top of a blue sphere behind a green pyramid", a higher CFG value leads to more semantically compliant images.

These findings prompt a fundamental rethinking of conditional generation tasks with strong conditions, such as aligning to various reward functions in text-to-image generation. For these tasks, we usually model the conditional density $p(x|y)$ through the decomposition $p(x|y) \propto p(x) \cdot p(y|x)$, where the terms correspond to the marginal density and discriminative model respectively. In diffusion

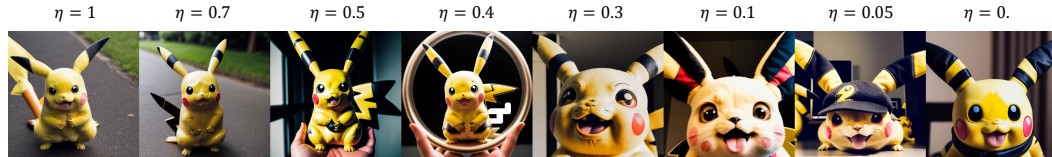

| $\eta = 1$ | $\eta = 0.7$ | $\eta = 0.5$ | $\eta = 0.4$ | $\eta = 0.3$ | $\eta = 0.1$ | $\eta = 0.05$ | $\eta = 0.$ |

Figure 4: Samples with 100 NFE and 7.5 CFG by varying the $\eta$ in sampling. The samples are generated from the same initial noise.

models, learning the marginal density $p(\boldsymbol{x})$ is usually the main focus while the condition part is usually handled by CFG or external guidance in diffusion models [2, 33]. However, as conditions get stronger and more complicated, the conditional distribution may be *ill-conditioned* to estimate, since the sample size for each condition is oftentimes only 1.

This leads us to the key insight: for tasks with strong conditions, there is a fundamental shift in the nature of the problem — moving away from directly estimating the conditional distribution towards what is more accurately described as **regularized rewards maximization**, where the discriminative part $p(y|\boldsymbol{x})$ captured by various rewards becomes the primary driving force, and the base generative model acts as a regularizer. In the following section, we will formally reformulate the problem of conditional generation with rewards.

### 3.2 Regularized Reward Maximization for Image Generation

Let $R(\boldsymbol{x})$ denote a reward function mapping $\mathbb{R}^d \to \mathbb{R}$. We can characterize the reward-centric generation task as searching in the *image domain* for samples that maximize rewards, which is naturally an optimization problem. To characterize the image domain constraint, consider having access to the image likelihood $p(\boldsymbol{x})$ for natural images. If $p(\boldsymbol{x})$ is below a certain threshold, $\boldsymbol{x}$ can be deemed outside the image domain. Therefore, we can rethink the generation as a regularized reward maximization problem with the following objective:

$$\max_{\boldsymbol{x} \in \mathbb{R}^d} R(\boldsymbol{x}) \text{ s.t. } p(\boldsymbol{x}) \leq c. \tag{3.1}$$

where $c > 0$ is some threshold. With this formulation, there are two important design choices to consider: (1) How to parameterize images? (2) How to effectively optimize the generator while preventing reward hacking? We will demonstrate in later sections that with a well-parameterized generator and proper regularization, we can achieve state-of-the-art text-to-image generation with only a few reward functions, without relying on diffusion distillation.

#### 3.2.1 Parameterizing Images with Generator

To achieve fast image generation, a natural strategy is to utilize GAN-style generators that directly transform random noise into images. With this formulation, the optimization problem shifts from the image space to the generator's parameter space. Ideally, the generator should have large-enough capacity and a reasonable initial grasp of the image distribution. We address this by proposing the use of an unrolled few-step sampler directly from pre-trained diffusion models. Conveniently, the number of unrolled steps allows us to control both the generator's capacity and its initial knowledge.

Specifically, we parameterize the generator $g_{\theta,\eta}$ to accept $K$ noisy levels of a diffusion model as inputs, creating a network that progresses from noise to obtain clean samples in $K$ steps. The specific parameterization is as follows:

$$g_{\theta,\eta}(\boldsymbol{z}) = g_{\theta,\sigma_1,\eta} \circ g_{\theta,\sigma_2,\eta} ... \circ g_{\theta,\sigma_K,\eta}(\boldsymbol{z}), \ g_{\theta,\sigma_k,\eta}(\boldsymbol{x}_k) = \sqrt{1 - \sigma_{k-1}^2} \frac{\boldsymbol{x}_k - \sigma_k \epsilon_\theta(\boldsymbol{x}_k)}{\sqrt{1 - \sigma_k^2}} + \sigma_{k-1} \widehat{\epsilon}_\eta \quad (3.2)$$

where $\widehat{\epsilon}_\eta = \eta \epsilon_\theta(\boldsymbol{x}_k) + \sqrt{1 - \eta^2} \epsilon$, $\sigma_K := 1$ and $\boldsymbol{z}, \epsilon \sim \mathcal{N}(0, I)$. We can use a pre-trained score net as the initialization for enough model capacity and better initial images. Our optimization target is $\theta$, which can be the full score network parameters or low-rank adaptation of them (LoRA) [20].

**Random $\eta$-Sampling.** The $\eta$ plays an important role in our parameterized generator $g_{\theta,\eta}$. In particular, when $\eta = 1$, the generator is parameterized into the discrete format of DDIM sampler [54]. In practice, we find that varying $\eta$ in diffusion sampling results in significant differences in the style

and layout of the generated images, as shown in Fig. 4. Motivated by this, we propose to parameterize our generator by inputting random $\eta$ at each step for augmenting the generator distribution that allows the generator explore more diverse area:

$$g_{\theta,\boldsymbol{\eta}}(\boldsymbol{z}) = g_{\theta,\sigma_1,\eta_1} \circ g_{\theta,\sigma_2,\eta_2}... \circ g_{\theta,\sigma_K,\eta_K}(\boldsymbol{z}), \tag{3.3}$$

where $\eta_i \sim U[0,1]$ and $i = 1, 2, .., K$. After training, we can fix a $\eta \in [0,1]$ in sampling. The design can effectively augment the generator distribution by randomly adding stochastic, serving as an effective regularization to enhance performance.

### 3.2.2 Optimization as Sampling from Reward-tilted Distribution

The generator $g_{\theta,\boldsymbol{\eta}}(\boldsymbol{z})$ initialized from the base diffusion model provides a non-trivial starting point for reward-centric training. We aim to adjust the parameter $\theta$ such that generated images are more preferred by the rewards. Formally, we can define the target $\theta^*$ to be distributed following a reward-informed posterior distribution

$$p^*(\theta) \propto p_0(\theta) \exp\{\bar{R}(\theta)\},$$

where $\bar{R}(\theta) = \mathbb{E}_{\boldsymbol{z},\boldsymbol{\eta}}[R(g_{\theta,\boldsymbol{\eta}}(\boldsymbol{z}))]$ and $p_0(\theta)$ is the initial distribution from the base diffusion models, which can be viewed as the likelihood to specify the image domain likelihood. For simplicity, we choose $p_0(\theta)$ to be a Gaussian distribution centered in the pretrained $\theta_0$ with variance $\sigma^2 I$. More advanced prior distributions will definitely lead to better performance.

**Remark 3.1.** The above formulation is similar to existing works in reward-driven approaches [40, 39, 51, 44, 8]. Given a base generative model with base distribution $p_0(\boldsymbol{x})$, the reward-tilted target is usually defined as $p^*(\boldsymbol{x}) \propto p_0(\boldsymbol{x}) \exp\{R(\boldsymbol{x})\}$. The key difference is that our formulation is on the generator parameter space.

With the target distribution $p^*$ specified, a straight-forward method to obtain $\theta^* \sim p^*(\theta)$ using stochastic Langevin dynamics. Concretely, starting from $\theta^{(0)} = \theta_0$, $\theta^{(t+1)}$ can be iteratively calculated via the stochastic gradient Langevin dynamics (SGLD) [59]

$$\theta^{(t+1)} - \theta^{(t)} = \lambda \nabla \log \left( p_0(\theta^{(t)}) \exp\{\bar{R}(\theta^{(t)})\} \right) + \sqrt{2\lambda}\epsilon_t$$

$$= \underbrace{\lambda \nabla \bar{R}(\theta^{(t)})}_{\substack{\text{Rewards} \\ \text{maximization}}} - \underbrace{\frac{\lambda}{2\sigma^2} \nabla \|\theta^{(t)} - \theta_0\|_2^2 + \sqrt{2\lambda}\epsilon_t}_{\text{Regularization}}, \tag{3.4}$$

where $\lambda$ is the learning rate and $\epsilon_t \sim N(0,1)$ is a random noise. As can be seen above, each update constitutes of reward maximization and regularization. The regularization is also two-fold, one being an $l_2$-penalty or weight decay, the other one being random noise perturbation.

### 3.3 Elucidating the Design Space of Reward-Instruct

By incorporating all aforementioned designs, we have developed a surprisingly simple yet effective reward-centric approach to fast image generation. Rooted in regularized reward maximization, we call our method Reward-Instruct and it is a framework that directly converts pre-trained base diffusion models into reward-enhanced few-step generators. Detailed algorithms are summarized in Algorithm 1 in the Appendix.

Such a simple reward-centric design significantly improves the training efficiency, training stability, resulting in state-of-the-art results in few-step text-to-image generation. Fig. 5 illustrates the qualitative comparisons among other methods. Detailed comparison for computation cost are deferred to Appendix B.1. Next, We proceed to explore the various facets of its design space.

### 3.3.1 Form of Regularization

The primary role of regularization is to constrain the generator's distribution to the vicinity of the image manifold, thus avoiding reward hacking. In our formulation, the specific form of the $\theta$-regularization is dependent on the form of $p_0(\theta)$. Choosing Gaussian distribution will give rise to $l_2$ regularization. This is very similar to the KL penalty in RL to control the update to be not too large [36]. The random noises introduced in the SGLD sampling algorithm also provide a regularization effect against reward hacking. Its effect is ablated in Appendix B.2.

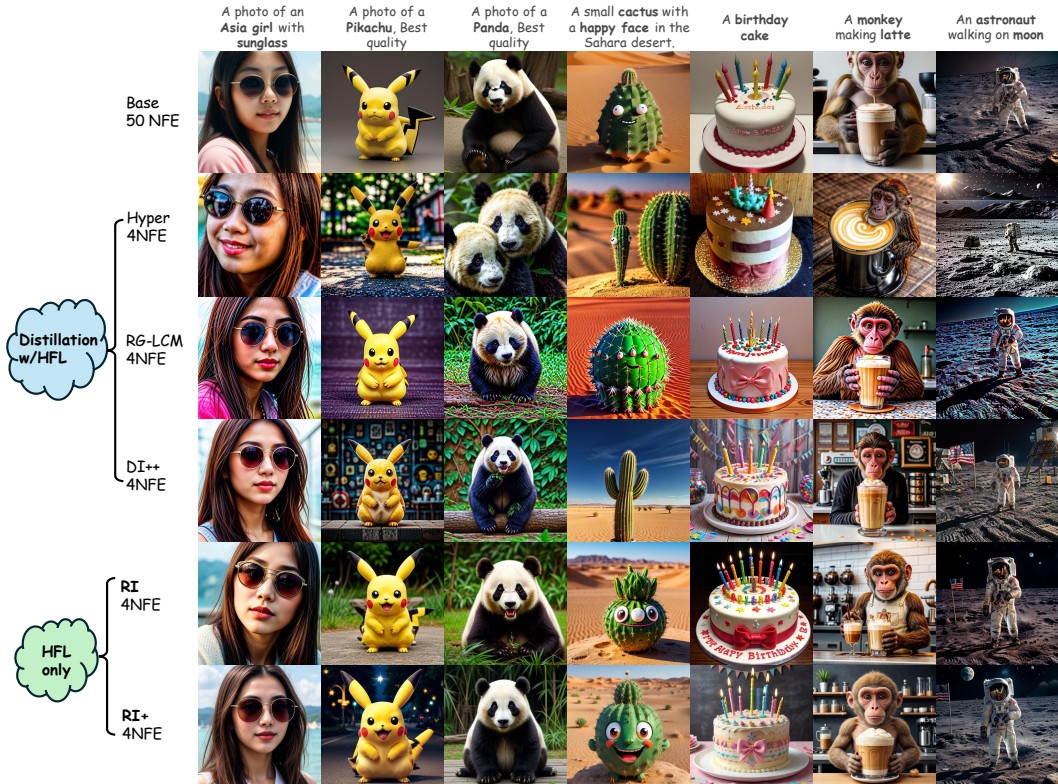

Figure 5: Qualitative comparisons of Reward-Instruct against distillation based and diffusion based models on SD-v1.5 backbone. All images are generated by the same initial noise. We surprisingly observe that our proposed Reward-Instruct has better image quality and text-image alignment compared to prior distillation-based reward maximization methods in 4-step text-to-image generation.

**Diffusion distillation as regularization.** More closely related to our generator parameterization, existing works such as RG-LCM [23] and DI++ [27] introduce reward maximization into diffusion distillation. These methods typically require training an additional score model for the distilled generator to ensure its closeness to the original model, which is memory and computation-intensive. Surprisingly, we found that the reward gradient dominates throughout the training process, turning diffusion distillation into a sort of costly regularization role (Fig. 3). Specifically, the RG-LCM and DI++ employ HPS v2.0 [61] or Image Reward [62] as the reward loss in their original approach, but we discovered that if the reward is set to HPS v2.1, it would cause the generators of RG-LCM and DI++ to collapse into undesirable distributions, as shown in Fig. 6.

Built upon the above formulation and regularization techniques, we found that our method integrated with HPS v2.1 can generate high-quality images, suffering less from artifacts compared to RG-LCM and DI++ (Fig. 7), where each regularization effectively improves image quality. This indicates the importance of proper regularization in reward maximization. However, we found that optimization with a single explicit reward still suffers from the over-saturation issue. To address this, we suggest maximizing multiple explicit rewards. This indicates the importance of proper regularization in reward maximization.

### 3.3.2 Power of Multiple Rewards

Thanks to a plethora of preference data and powerful RL methods, we have access to a diverse collection of learned reward models. Although learning from a zoo of pre-trained models has long been studied in various vision tasks [60, 9, 4, 10], utilizing multiple reward models is relatively underexplored. Denote $R_1(\boldsymbol{x}), \ldots, R_m(\boldsymbol{x})$ as the reward functions, each with its own set of modes, some genuinely good and some corresponding to artifacts. Our assumption is that the good ones are associated with the ground truth, while the bad ones are random and not shared with other reward

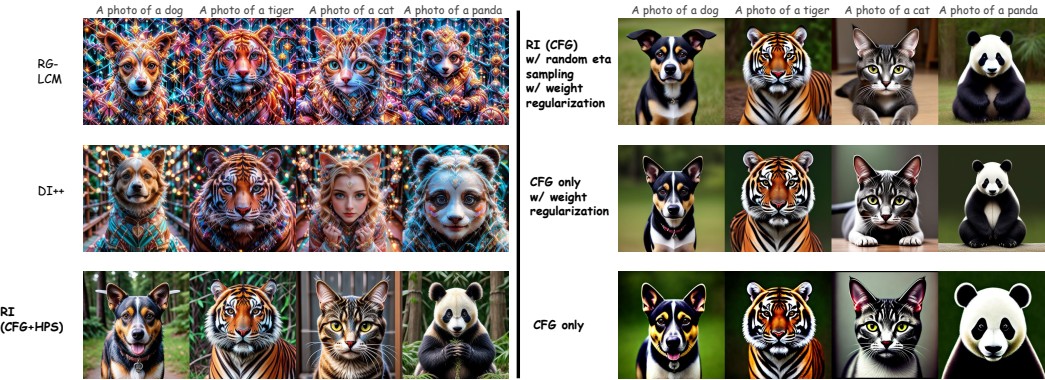

Figure 6: The prior distillation-based reward maximization methods collapse when the reward is chosen to be HPS v2.1. In contrast, our Reward-Instruct still works well, benefiting from the proposed effective regularization technique.

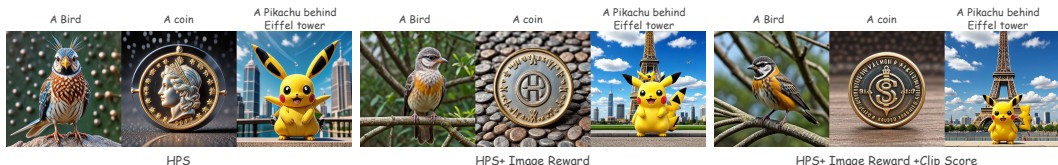

Figure 7: Four-step samples generated by Reward-Instruct. We observed that the quality of the image monotonically increases with the gradual increment of the reward count.

functions. Therefore, with a diverse collection of different reward functions focusing on different aspects of the data, images preferred by all of them tend to be good.

Therefore, the goal of utilizing multiple rewards is to find common data modes that have high rewards from different perspectives. This design can effectively prevent the model from hacking each individual reward, therefore significantly stabilizing the training process. We can extend (3.1) to be

$$\max_{\boldsymbol{x} \in \mathbb{R}^d} \sum_{i=1}^{m} \omega_i R_i(\boldsymbol{x}) \text{ s.t. } p(\boldsymbol{x}) \leq c, \tag{3.5}$$

where $\omega_i$'s are positive weights for balancing the rewards. However, all we have is a gradient pointing towards the direction of the steepest climb. How to find the most effective direction is of critical importance. In practice, we found that optimizing multiple rewards with a naive *weighted combination* may fail to maximize all rewards in the training process. As shown in Fig. 9, the clip score does not converge to a high value.

To balance the learning of different rewards, we suggest *gradient normalization*, normalizing the gradients from each reward and forming the average direction. Therefore, we can choose the weights $\omega_i$ in (3.1) as

$$\omega_i = \widehat{\omega}_i / \text{sg}(||\frac{\partial R_i(g(\boldsymbol{z}))}{\partial g(\boldsymbol{z})}||_2). \tag{3.6}$$

This is equal to setting a dynamic weighting. Note that the normalizing operation is also performed for the implicit reward (CFG). Fig. 9 shows that after applying the gradient normalization, we can maximize multiple rewards well. Fig. 7 demonstrates that the image quality and image-text alignment become significantly better when we maximize multiple rewards.

**The Complementary Effect of Rewards.** Since the common modes of multiple rewards tend to be more well-behaved than those from signal rewards, the combination of multiple rewards in (3.6) also serves as a kind of implicit regularization for the generator. To verify this, we maximize HPS, image reward, and clip score separately, without using random-$\eta$ sampling. We find that individual rewards perform poorly, but when combined, they can generate images of reasonable quality as shown in Fig. 8, which highly emphasizes the complementarity between rewards and their effectiveness as implicit regularization.

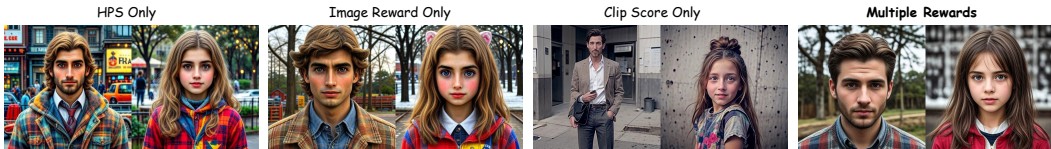

Figure 8: The **complementary effect** of different reward. We do not use random $\eta$ sampling and set small weight regularization in training here to highlight the complementary effect between rewards.

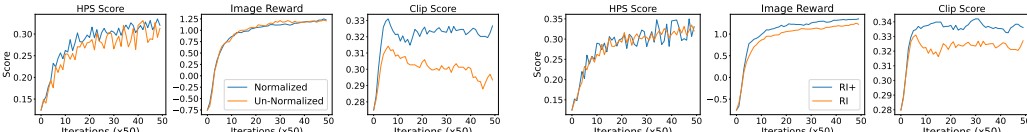

Figure 9: Training progress of various metrics over iterations. It can be seen that the normalized gradient shows better performance. This is evaluated on 1k prompts from HPS benchmark.

Figure 10: Comparison on the convergence speed between RI and RI+. We note that each training iteration of RI+ takes only 65% of the time required by RI.

### 3.4 Reward-Instruct+: Additional Supervision on Intermediate Steps

The reward supervision in Reward-Instruct is only at the final generated samples in an end-to-end fashion, which is similar to the DeepSeek-R1 [7]. To further enhance performance, we can incorporate extra supervision into intermediate generation steps to form Reward-Instruct+.

Although Reward-Instruct is already capable of generating high-quality images in a setting where only reward maximization is considered, the reward signal is only provided at the end. This leads to two issues: on one hand, the efficiency of reward maximization is low because there is no direct reward signal feedback during the process; on the other hand, the gradient needs to be backpropagated through the entire generator, resulting in high memory usage and significant computational costs.

To address the above issues, we propose learning the generator with intermediate supervision as well, forming Reward-Instruct+. We rewrite the generator as follows:

$$
\boldsymbol{x}_{k+1} = \mathrm{sg}(g_{\theta,\sigma_{k+2},\eta_{k+2}} \circ g_{\theta,\sigma_{k+3},\eta_{k+2}}... \circ g_{\theta,\sigma_K,\eta_K}(\boldsymbol{z})),
$$
$$
\boldsymbol{x}_k = \sqrt{1-\sigma_k^2}\frac{\boldsymbol{x}_{k+1} - \sigma_{k+1}\epsilon_\theta(\boldsymbol{x}_{k+1})}{\sqrt{1-\sigma_{k+1}^2}} + \sigma_k\widehat{\epsilon}_{\eta_k}, \quad \boldsymbol{x}_0^{(k)} = \frac{\boldsymbol{x}_{k+1} - \sigma_{k+1}\epsilon_\theta(\boldsymbol{x}_{k+1})}{\sqrt{1-\sigma_{k+1}^2}} \tag{3.7}
$$

where $k = 1, 2, ..., K$, $\eta_i \sim U[0,1]$, $\widehat{\epsilon}_{\eta_k} = \eta_k\epsilon_\theta(\boldsymbol{x}_{k+1}) + \sqrt{1-\eta_k^2}\epsilon$, $\sigma_K := 1$ and $\boldsymbol{z}, \epsilon \sim \mathcal{N}(0, I)$. For computing CFG, we diffuse samples from $\boldsymbol{x}_k$. The $k$ is randomly sampled during training. The update of $\theta$ has the same form as Eq. (3.4), differing in how to obtain samples.

By doing so, we can effectively supervise the intermediate process in the generation process. Fig. 10 shows that after applying the intermediate supervision, the convergence speed of Reward-Instruct+ has significantly improved compared to Reward-Instruct. Moreover, training RI+ is more efficient than RI, since each training iteration of RI+ takes only 65% of the time required by RI.

**Comparison on Generation Path Between Reward-Instruct and Reward-Instruct+.** Reward-Instruct and Reward-Instruct+ are significantly different and it is interesting to explore the generation paths corresponding to these two models. In Fig. 16, we can observe that Reward-Instruct+ generates much clearer images during the early process compared to Reward-Instruct. Although Reward-Instruct+ struggles with severe artifacts in the early stages of generation, surprisingly, these artifacts are gradually removed rather than accumulating throughout the process. In contrast, Reward-Instruct's path progresses from blurry to clear, with less affected by artifacts during the process.

## 4 Evaluations

To verify the effectiveness of the Reward-Instruct and Reward-Instruct+, we compare them with previous distillation-based reward maximization methods. We put the experiment details in the Appendix.

Table 1: Comparison of machine metrics on text-to-image generation across SOTA methods. We **highlight** the best among fast sampling methods. The FID is measured based on COCO-5k dataset.

| Model | Type | Backbone | NFE | HPS↑ | Aes↑ | CS↑ | FID↓ | Image Reward↑ |
|-------|------|----------|-----|------|------|-----|------|---------------|
| Base Model (Realistic-vision) | | SD-v1.5 | 50 | 30.19 | 5.87 | 34.28 | 29.11 | 0.81 |
| Hyper-SD [45] | | SD-v1.5 | 4 | 30.24 | 5.78 | 31.49 | **30.32** | 0.90 |
| RG-LCM [23] | Distillation + Reward | SD-v1.5 | 4 | 31.44 | 6.12 | 29.14 | 52.01 | 0.67 |
| DI++ [27] | | SD-v1.5 | 4 | 31.83 | 6.09 | 29.22 | 55.52 | 0.72 |
| ReFL [62] | Reward Centric | SD-v1.5 | 50 | 31.82 | 5.97 | 31.78 | 39.38 | 1.16 |
| DRaFT [5] | | SD-v1.5 | 50 | 33.10 | 6.18 | 30.70 | 37.10 | 0.85 |
| Reward-Instruct (Ours) | Reward Centric | SD-v1.5 | 4 | 33.70 | 6.11 | 32.13 | 33.79 | 1.22 |
| Reward-Instruct+ (Ours) | | SD-v1.5 | 4 | **34.37** | **6.20** | **32.97** | 37.53 | **1.27** |
| Base Model | | SD3-Medium | 56 | 31.37 | 5.84 | 34.13 | 28.72 | 1.07 |
| Reward-Instruct (Ours) | Reward Centric | SD3-Medium | 4 | 34.04 | 6.27 | 33.89 | 31.97 | 1.13 |

w/ Dreamshaper   Image Editing: squirrel -> cat   w/ ControlNet: Canny -> Image

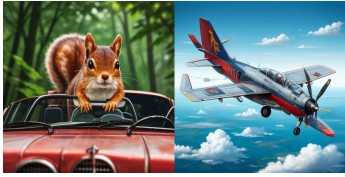 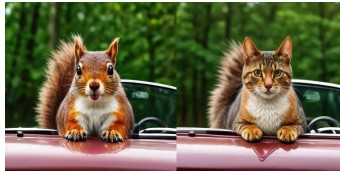 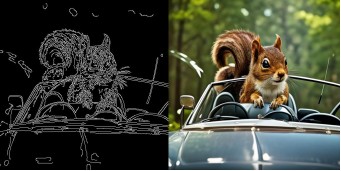

Figure 11: Qualitative comparison against competing methods and applications in downstream tasks.

Table 3: Ablation study on the proposed components in our Reward-Instruct.

| Model | Backbone | Steps | HPS↑ | Aes↑ | CS↑ | FID↓ | Image Reward↑ |
|-------|----------|-------|------|------|-----|------|---------------|
| **Reward-Instruct (Ours)** | SD-v1.5 | 4 | 33.70 | 6.11 | 32.13 | 33.79 | 1.22 |
| Reward-Instruct w/ single reward | SD-v1.5 | 4 | 32.08 | 5.80 | 31.01 | 38.21 | 0.89 |
| Reward-Instruct w/o random-$\eta$-sampling | SD-v1.5 | 4 | 33.41 | 6.12 | 32.11 | 36.90 | 1.18 |
| Reward-Instruct w/o weight regularization | SD-v1.5 | 4 | 34.26 | 6.12 | 32.45 | 39.25 | 1.27 |

**Metric.** We assess image quality using the Aesthetic Score (AeS) [50], image-text alignment and human preference with the Human Preference Score (HPS) v2.1 [61] and Image Reward, and image-text alignment with the CLIP score (CS) [16]. Additionally, we use zero-shot FID on the COCO-5k dataset for a more comprehensive evaluation.

**Qualitative Results.** We present the qualitative results in Fig. 5. It can be observed that our proposed Reward-Instruct and Reward-Instruct+, without using a distillation loss, demonstrate better image quality and text-image alignment compared to existing distillation-based reward maximization methods and RL-based finetuning methods. We also include additional visualization regarding complex prompts in Fig. 15.

**User Study.** To further verify the effectiveness of our proposed method without concern for reward-hacking, we conduct a user study on SD-v1.5 backbone. We refer to Section B.2 for details of the user study. The results in Table 2 show that our method outperforms the base diffusion model.

Table 2: User Preference Study.

| Model | NFE | User Preference↑ |
|-------|-----|------------------|
| Base Model | 50 | 41.7% |
| RI (Ours) | 4 | **58.3%** |

**Quantitative Results.** We present the quantitative results in Table 1. Our proposed Reward-Instruct and Reward-Instruct+ achieve state-of-the-art (SOTA) performance across various text-image alignment and human preference metrics. Notably, our model also achieves a zero-shot COCO FID comparable to the original model, which demonstrates that our method does not suffer from artifacts.

**Ablation Study.** We provide quantitative results about removing one technique at a time to show the effectiveness. The results in Table 3 demonstrate the critical role of each component: 1) Multiple rewards: Improves both rewards and FID, mitigating artifacts and reward hacking. 2) Random-sampling: Maintains similar reward performance but significantly improves FID, aiding to find better mode with fewer artifacts. 3) Weight regularization: Trades slight reward gains for better FID, ensuring the generator stays within the image manifold.

**Additional Application.** We show our Reward-Instruct's capabilities in various tasks: **1) Image-to-Image Editing:** As shown in Fig. 11, Reward-Instruct performs high-quality image editing [35] in

just four steps; **2) Compatibility with ControlNet and Base Models:** Illustrated in Fig. 11, Reward-Instruct-LoRA is compatible with ControlNet [69] and works seamlessly with various fine-tuned base models (e.g., Dreamshaper from SD 1.5), preserving their unique styles.

## 5   Discussion

Our work targets a reward-centric approach to photo-realistic image generation. Our proposed Reward-Instruct demonstrates that via regularized reward maximization, we can convert pretrained base diffusion models to reward-aligned few-step generators, without diffusion distillation losses or training images. Our method enables 4-step 1024px generation (Fig. 1), matching or exceeding both the inference speed and sample quality of previous approaches. In Appendix C, we further explore more application scenarios and extensions where our method can shine.

Our approach is inspiring but have its limitations. For instance, our method relies on trained differentiable reward functions. It's an interesting future work to extend such a reward-centric approach to include black-box reward functions. It would be exciting to explore DPO-style [44] pipelines that directly fine-tune generators with raw preference data, and extending these principles to other generative domains like video and 3D content. Moreover, our Reward-Instruct framework can potentially work with any few-step generator and doesn't have to relate to diffusion models but this is not explored in the current work. Making this investigating is an important extension.

## Acknowledgments

Jing Tang's work is partially supported by National Key R&D Program of China under Grant No. 2024YFA1012700 and No. 2023YFF0725100, by the National Natural Science Foundation of China (NSFC) under Grant No. 62402410 and No. U22B2060, by Guangdong Provincial Project (No. 2023QN10X025), by Guangdong Basic and Applied Basic Research Foundation under Grant No. 2023A1515110131, by Guangzhou Municipal Science and Technology Bureau under Grant No. 2024A04J4454, by Guangzhou Municipal Education Bureau (No. 2024312263), and by Guangzhou Industrial Information and Intelligent Key Laboratory Project (No. 2024A03J0628) and Guangzhou Municipal Key Laboratory of Financial Technology Cutting-Edge Research (No. 2024A03J0630).

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

# A   Additional Related Works

**Few-Step Diffusion Sampling.** Despite significant advancements in training-free accelerated sampling of DMs [25, 71, 64, 52, 34], diffusion distillation [26] is dispensable for satisfactory few-step Sampling. Typically, the distilled sampler involves a single or multiple transformations from random noise to images. Among various approaches, trajectory matching [56, 21, 55, 15, 48] and distribution matching [68, 28, 72, 49, 63, 30] are the most popular methods for diffusion distillation in few-step diffusion sampling. Very recently, trajectory distribution matching [32] has shown its promising performance in distillation.

**Preference Alignment for Text-to-Image Models.** In recent years, significant efforts have been made to align diffusion models with human preferences. These approaches can be broadly categorized into three main lines of work: 1) fine-tuning DMs on carefully curated image-prompt datasets [6, 41]; 2) maximizing explicit reward functions, either through multi-step diffusion generation outputs [43, 5, 22, 17] or policy gradient-based reinforcement learning (RL) methods [14, 3, 67]. 3) implicit reward maximization, exemplified by Diffusion-DPO [58] and Diffusion-KTO [66], directly utilizes raw preference data without the need for explicit reward functions.

**Reward driven image generation.** There is an active line of research investigating using various rewards in the post-training process for improved image alignment. For instance, [13] proposed to fine-tune DDPM samplers via policy gradient, achieving good few-step sampling performance. [8] considered reward fine-tuning diffusion models or flow models via stochastic optimal control (SOC), and proposed Adjoint Matching which outperforms existing SOC algorithms. ReNO [12] optimizes the initial noise by maximizing multiple rewards for enhanced performance given a frozen one-step generator. Our Reward-Instruct significantly differs from existing works in that it is the first reward-driven few-step image generation method that directly converts pretrained diffusion models to reward-enhanced few-step generators, without relying on complicated diffusion distillation or training data.

Another closely related work is "Referee can play" [24], where the authors emphasized the importance of discriminative models in conditional generation and presented a text-to-image generation pipeline by inverting Vision Language Models (VLMs). Specifically, they utilized the decoder from stable diffusion and optimized the latent for maximizing the alignment score given by VLMs. On a high-level, the VLMs employed in [24] can also be viewed as reward models, providing matching scores for text-image pairs. Even though [24] provides an interesting proof-of-concept demonstration, it lacks a systematic formulation and its generating process has major downsides in real-world scenarios. (1) The optimization process is highly sensitive to initialization and hyperparameter choices, which is not robust. (2) To generate a new image, they have to do hundreds of function evaluations (NFEs), which is computationally intensive. Visual comparisons are illustrated in Figure 12.

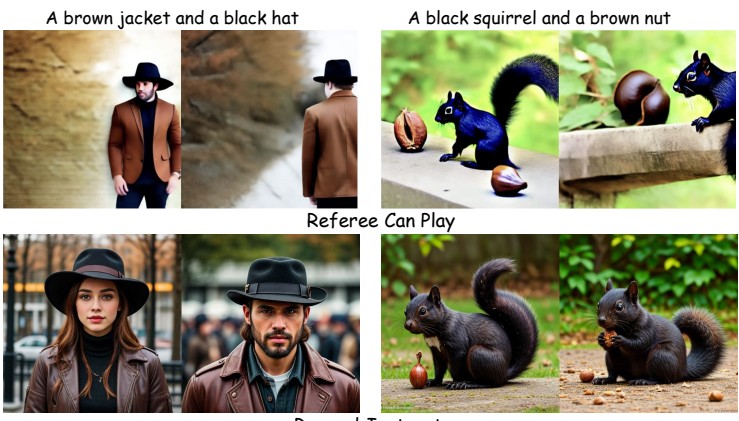

Figure 12: Comparison with **Referee can play** [24]. The baseline samples are taken from their paper. It can be seen that our Reward-Instruct has significantly better visual quality.

---
**Algorithm 1** Reward-Instruct
---

**Require:** Generator $f_\theta$, Pre-trained score $f_\psi$, Reward models $\{r_i\}$, desired sampling steps $K$, total iterations $N$, learning rate $\lambda$.

**Ensure:** optimized generator $f_\theta$.

1: Initialize weights $\theta$ by $\psi$;
2: **for** $i \leftarrow 1$ **to** $N$ **do**
3:     Sample noise $\epsilon$ from standard normal distribution;
4:     Sample $x$ with initialized noise $\epsilon$ from generator $f_\theta$ by $K$ steps via random-$\eta$ sampling.
5:     # Compute regularization loss
6:     $\mathcal{L}_{reg} \leftarrow \|\theta - \psi\|_2^2$
7:     # Compute Rewards
8:     $\mathcal{L}_{reward} \leftarrow -\sum_i \frac{\widehat{\omega}_i}{\text{sg}(||\frac{\partial R_i(\boldsymbol{x})}{\partial \boldsymbol{x}}||_2)} R_i(\boldsymbol{x}, y)$
9:     # Compute CFG guidance loss
10:    Sample $\boldsymbol{x}_t \sim q(\boldsymbol{x}_t|\boldsymbol{x})$
11:    cfg_grad $\leftarrow \nabla_{\boldsymbol{x}_t} \log p_\psi(c|\boldsymbol{x}_t)$
12:    $\mathcal{L}_{cfg} \leftarrow \|\boldsymbol{x}_t - \text{sg}(\boldsymbol{x}_t + \text{cfg\_grad})\|_2^2$
13:    # Compute Total Loss and Update
14:    $\mathcal{L}_{total} \leftarrow \omega_{reg}\mathcal{L}_{reg} + \mathcal{L}_{reward} + \omega_{cfg}\mathcal{L}_{cfg}$
15:    Update $\theta$ using SGLD step with temperature $\tau$: $\theta \leftarrow \theta - \frac{\lambda}{\tau}\nabla_\theta \mathcal{L}_{total} + \sqrt{2\lambda}\epsilon$.
16: **end for**

---

Listing 1: Torch-style pseudo code of SGLD step with temperature $\tau$.

```
loss.backward()
noise_scale = (2 * learning_rate * tau) ** 0.5 / learning_rate
for param in model.parameters():
    with torch.no_grad():
        if param.grad is not None:
            noise_para = torch.randn_like(param.grad.data) *
                noise_scale
            param.grad.data.add_(noise_para)
optimizer.step()
optimizer.zero_grad()
```

## B   Experiment Details

**Baseline Models.** We perform experiments on SD-v1.5 [46] and SD3-medium [11], including both UNet and MM-DiT [11, 38] architectures, indicating the broad applicability of our approach.

**Experiment Setting.** Training is performed on the JourneyDB dataset [37] using prompts, without requiring images, as our method is image-free. We primarily compare with previous distillation-based reward maximization methods. For Hyper-SD, we use the public checkpoint, while for RG-LCM and DI++, we reproduce their methods. We adopt the AdamW optimizer with $\beta_1 = 0.9$, $\beta_2 = 0.95$, and the learning rate of $2e-5$. We use a batch size of 256.

**Details of User Study.** We randomly selected 20 prompts for image generation. Around 20 user responses are collected on 20 pairs in total. We note that multiple users review the same pair of images, which potentially reduces evaluation bias.

### B.1   Training Efficiency of Reward-Instruct

Our proposed Reward-Instruct is efficient to train, since it does not require training images and does not require online auxiliary models during training. And its overall training cost is also not expensive compared to multi-step RL-based methods and distillation-based RL methods, as shown in the Table 4.

| Method | Few-step | Image-Free | Single Trainable Network | Training Cost |
|---|---|---|---|---|
| DRaFT | ✗ | ✓ | ✓ | 72 Hours |
| ReFL | ✗ | ✓ | ✓ | 64 Hours |
| RG-LCM | ✓ | ✗ | ✓ | 20 Hours |
| DI++ | ✓ | ✓ | ✗ | 36 Hours |
| Reward-Instruct (Ours) | ✓ | ✓ | ✓ | 28 Hours |

Table 4: Comparison on the training efficiency. Our method is efficient for training and deploying in various aspects. The training cost is measured by GPU hours on RTX-4090. We use the same batch size and iterations for each method.

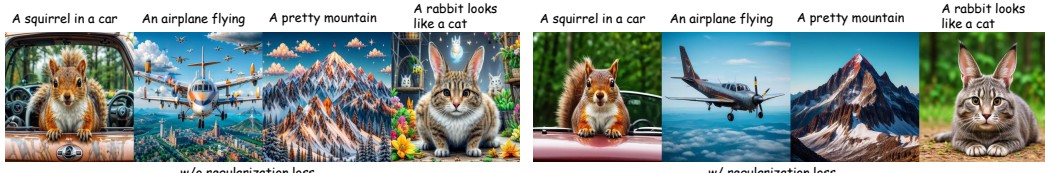

Figure 13: The effect of regularization loss. Images are from the same initial noise.

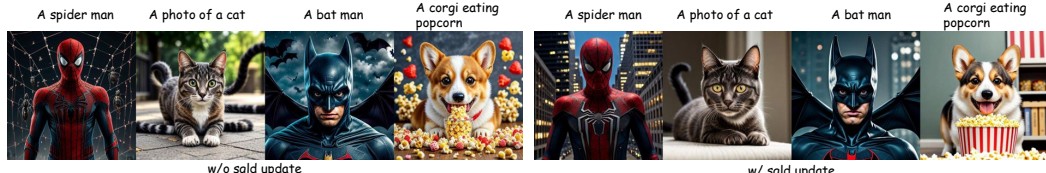

Figure 14: The effect of SGLD update. Images are from the same initial noise.

Table 5: Comparison between Reward-Instruct using different rewards.

| Model | Backbone | Steps | HPS↑ | Aes↑ | CS↑ | FID↓ | Image Reward↑ |
|---|---|---|---|---|---|---|---|
| Reward-Instruct (Ours) | SD-v1.5 | 4 | 33.70 | 6.11 | 32.13 | 33.79 | 1.22 |
| Reward-Instruct w/ HPS + Clip Score + AeS (Ours) | SD-v1.5 | 4 | 32.35 | 6.21 | 32.43 | 31.26 | 1.01 |

## B.2 Additional Ablation

**The Indispensability of Weight Regularization.** The regularization loss is the key to learning a good generator. Without weight regularization, even if we employ multiple rewards, the generator can easily converge to an undesirable distribution, which underscores the importance of explicitly regularizing the generator, as shown in Fig. 13.

**The Indispensability of SGLD's randomness.** The SGLD randomness also serves as a key to regularize the generator. We observe that by introducing SGLD's randomness in learning, the model is more robust to avoid artifacts. Under larger reward weightings, without SGLD's randomness, the model may generate some repeated objects in the background related to the prompts, as shown in Fig. 14.

**Different Rewards for Training Reward-Instruct.** We provided additional experiments incorporating Aesthetic Score (AeS) in Table 5. The results demonstrate that incorporating AeS — a reward focused solely on image aesthetics — maintains strong performance and even improves the zero-shot FID (from 33.79 to 31.26). This improvement may stem from AeS's complementary effect, as it emphasizes visual aesthetics differently from HPS and CLIP, potentially enhancing the regularization effect. These findings suggest that other rewards can indeed be effective to train our Reward-Instruct.

**Verify the Complementarity of Rewards.** We can assess the complementarity of the rewards to some extent by examining their correlations on a validation set. Specifically, we computed the Pearson correlation among the rewards used during training, utilizing the COCO5k dataset. The

|              | HPS  | Image Reward | Clip Score |
|--------------|------|--------------|------------|
| HPS          | 1    | 0.51         | 0.46       |
| Image Reward | 0.51 | 1            | 0.41       |
| Clip Score   | 0.46 | 0.41         | 1          |

Table 6: Pearson correlation between rewards computed on the COCO5k dataset

A stylish woman walks down a Tokyo street filled with warm glowing neon and animated city signage. She wears a black leather jacket, a long red dress, and black boots, and carries a black purse. She wears sunglasses and red lipstick. She walks confidently and casually. The street is damp and reflective, creating a mirror effect of the colorful lights. Many pedestrians walk about.

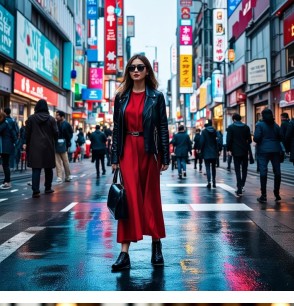

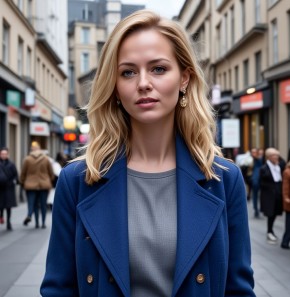

A close-up photo of a a woman. She wore a blue coat with a gray dress underneath. She has blue eyes and blond hair, and wears a pair of earrings.

An extreme close-up of an gray-haired man with a beard in his 60s, he is deep in thought pondering the history of the universe as he sits at a cafe in Paris, his eyes focus on people offscreen as they walk as he sits mostly motionless, he is dressed in a wool coat suit coat with a button-down shirt , he wears a brown beret and glasses and has a very professorial appearance.

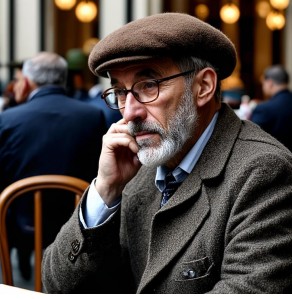

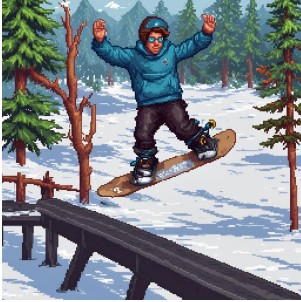

Pixel art style of a snowboarder in mid-air performs a trick on a black rail, wearing a blue sweatshirt and black pants, with arms outstretched. The serene snowy landscape background, dotted with trees, complements the scene.

Figure 15: Samples generated by Reward-Instruct (4 NFE) with long and complex prompts. The Reward-Instruct here is post-trained from `SD3-medium`.

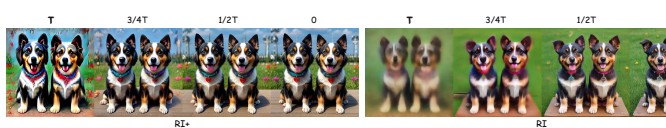

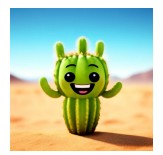

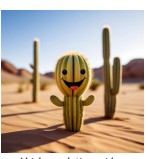

Figure 16: Path comparison between Reward-Instruct and Reward-Instruct+. The prompt is "Two dogs, best quality".

Figure 17: The effect of high-resolution guidance loss. Images are from the same noise.

results in Table 6 indicate that correlations among the different rewards are positive and relatively strong.

# C   More Applications for Reward-Instruct

## C.1   Super-resolution

**Tackling High-Resolution Generation.** Existing reward functions for text-to-image generation are mostly trained on low-resolution inputs (e.g., 224×224 pixels). This creates a fundamental limitation: directly optimizing such rewards during high-resolution (e.g., 1024×1024 pixels) synthesis struggles to preserve fine-grained details. To address this, we propose training a high-resolution classifier as a complementary guidance signal. This classifier explicitly prioritizes perceptual quality in high-resolution outputs.

**Implicit High-Resolution Classifier.** We can form a high-resolution classifier via the Bayesian rule:

$$\log p(\text{HighRes}|\boldsymbol{x}_t) = \log \frac{p(\boldsymbol{x}_t|\text{HighRes})p(\text{HighRes})}{p(\boldsymbol{x}_t|\text{LowRes})}, \tag{C.1}$$

where $\boldsymbol{x}_t$ denotes the noisy samples at timesteps $t$. Its gradient can be obtained as follows:

$$\nabla_{\boldsymbol{x}_t} \log p(\text{HighRes}|\boldsymbol{x}_t) = \nabla_{\boldsymbol{x}_t} \log(\boldsymbol{x}_t|\text{HighRes}) - \nabla_{\boldsymbol{x}_t} \log p(\boldsymbol{x}_t|\text{LowRes}). \tag{C.2}$$

The $\nabla_{\boldsymbol{x}_t} \log p(\boldsymbol{x}_t|\text{HighRes})$ can be directly replaced by the original diffusion model, since it has the capability to generate high-resolution images. For obtaining $\nabla_{\boldsymbol{x}_t} \log p(\boldsymbol{x}_t|\text{LowRes})$, we can finetune the pre-trained diffusion over low-resolution data. By doing so, we can obtain a powerful Implicit classifier for high-resolution guidance. Fig. 17 shows that after applying the implicit high-resolution guidance proposed by us, the generated images are significantly clearer.

## D  Limitations

Our model, akin to most text-to-image diffusion models, may not always perform perfectly regarding fairness and the accurate depiction of specific details. We intend to investigate these outstanding challenges within the generation field in our subsequent research. The goal of this future work will be to improve the model's proficiency in text generation, ensure fairer outcomes, and provide finer control over generated details.

## E  Broader Impacts

This work introduces Reward-Instruct, a reward-centric approach for developing efficient text-to-image models. From a positive perspective, although this is an academic study, we believe that the proposed Reward-Instruct could be widely adopted in industry. Conversely, if these swift generation models are exploited by individuals with malicious intent, they could also streamline and expedite the production of detrimental content. While our work is centered on scientific inquiry, we are committed to implementing measures to mitigate the spread of harmful information, such as by removing inappropriate content in the dataset.

## F  Safeguards

The Reward-Instruct is trained on the prompt of JourneyDB dataset [37], which has undergone rigorous human and machine-based filtering to ensure that there are no harmful or violent prompts in the dataset.

