# OpenReview forum: "Reward-Instruct: A Reward-Centric Approach to Fast Photo-Realistic Image Generation"
_NeurIPS.cc/2025/Conference — NeurIPS 2025 poster_

### Official Review · Reviewer_oEgS · 2025-06-22

**Clarity:** 3
**Significance:** 3
**Originality:** 3
**Rating:** 4
**Confidence:** 3

**Summary:**

The paper investigates the effect of rewards for fast image generation using pre-trained diffusion models.
The authors reformulate the problem of conditional generation with rewards, focusing on the reward and searching for samples that maximize it, while using the base generative model as a regularizer.
Additionally, they experiment with various reward functions and show that using a weighted normalized average of multiple rewards yields the best results.

**Questions:**

1. What are the results of Reward-Instruct over more complex prompts such as the ones presented in Figure 2?
2. “Figure 6: The prior distillation-based reward maximization methods collapse when the reward is chosen to be HPS v2.1.” – what happens for prior approaches vs. Reward-Instruct when not using HPS v2.1? (e.g. using HPS v2.0 [60] or Image Reward [61] as in their original approach?)
3. Figure 7 – there is a mild improvement when increasing the number of rewards used, but it seems more like enhancing the gradient step because the improvement is generally toward the same direction. My question is – what happens if you use a higher weight for one of the regularization methods instead of using the weighted average?
4. In addition to text proofreading that is needed, in some figures it is not clear what are the prompts for the examples, e.g. Figure 8, or it looks like the wrong prompt/images, e.g. in Figure 7 the "squirrel" example which shows bird images. These should be fixed.

**Ethical Concerns:**

["NO or VERY MINOR ethics concerns only"]

**Final Justification:**

The authors addressed most of my concerns by providing clarifications and additional experiments.
Assuming the authors will proofread the paper and include more complex prompts comparisons with competing methods, I lean toward acceptance.

**Limitations:**

I think the paper could benefit from a broader discussion of the method's limitations, including visual examples of them.

**Paper Formatting Concerns:**

No concerns.

**Quality:**

3

**Strengths And Weaknesses:**

Strengths:
* Clear background and motivation.
* The authors offer an interesting re-formulation of image generation formula post pre-training by placing the reward in the center.
* The evaluation is thorough and includes ablations and comparisons with relevant previous works.

Weaknesses:
* The paper needs proofreading as it contains many typos and grammatical issues, for example:
“Our journey starts with an analysis in Section 3 shows that” (line 57), “our methods directly operates..” (line 66), “advanced prior distribution will definitely leads to..” (line 166), “converts pretrained pre-trained base..” (line 180), and more..
* The limitations section is quite limited, I think the paper could benefit from further limitations discussion including images of failure cases.
* There aren't many visual examples, and there aren't complex visual examples of the method as the ones presented in Figure 2 of limitations of the competing method.

---

> ### Author Rebuttal · Authors · 2025-07-31
>
> We sincerely thank you for the valuable feedback. We address each comment below.
>
> > The paper needs proofreading as it contains many typos and grammatical issues
>
> Thank you for your valuable comments. We will revise those typos and grammatical issues in the revision.
>
> > The limitations section is quite limited, I think the paper could benefit from further limitations discussion including images of failure cases.
>
> Thank you for your great suggestion. We will discuss our limitations in greater detail in our revision. Specifically, we will add some failure cases and corresponding analysis.
>
> > What are the results of Reward-Instruct over more complex prompts such as the ones presented in Figure 2?
>
> Our model also performs well on complex prompts. For example, the prompts used in Figure 1 have some complex cases: "A alpaca made of colorful building blocks, realism", "A professional portrait photo of an anthropomorphic capybara wearing fancy gentleman hat and jacket walking in autumn forest". We will add more samples on complex prompts in revision, but we cannot provide it in the rebuttal stage due to the policy.
>
> > “Figure 6: The prior distillation-based reward maximization methods collapse when the reward is chosen to be HPS v2.1.” – what happens for prior approaches vs. Reward-Instruct when not using HPS v2.1? (e.g. using HPS v2.0 [60] or Image Reward [61] as in their original approach?)
>
> Great question. The results in Figure 5 and Table 1 for RG-LCM and DI++ are obtained by using the reward in their original approach. It can be seen that our proposed Reward-Instruct achieves better performance.
>
> > Figure 7 – there is a mild improvement when increasing the number of rewards used, but it seems more like enhancing the gradient step because the improvement is generally toward the same direction. My question is – what happens if you use a higher weight for one of the regularization methods instead of using the weighted average?
>
> We argue that the performance improvement brought by multiple rewards is not merely due to the enhancement of gradient step. We presented the results using weak regularization in Figure 8 (small weight regularization coefficient and without using random eta sampling), *equivalent to a higher weight of rewards*. From Figure 8, it can be observed that using a single reward leads to severe artifacts, while ensembling through multiple rewards can generate reasonably high-quality images.
>
> Besides, we conduct an extra study on using a large weight regularization coefficient (5 times larger). The results are shown below:
>
> | Model                      | Backbone | Steps | HPS↑  | Aes↑ | CS↑   | FID↓  | Image Reward↑ |
> | -------------------------- | -------- | ----- | ----- | ---- | ----- | ----- | ------------- |
> | R0 w/ single reward        | SD-v1.5  | 4     | 32.08 | 5.80 | 31.01 | 38.21 | 0.89          |
> | R0 w/ large regularization | SD-v1.5  | 4     | 32.86 | 5.98 | 31.50 | 32.86 | 0.95          |
> | R0 (Ours)                  | SD-v1.5  | 4     | 33.70 | 6.11 | 32.13 | 33.79 | 1.22          |
>
> It can be seen that with large regularization and multiple rewards, the performance on rewards is degraded while the zero-shot FID becomes better. And its performance is way better than using only single reward.
>
> > In addition to text proofreading that is needed, in some figures it is not clear what are the prompts for the examples, e.g. Figure 8, or it looks like the wrong prompt/images, e.g. in Figure 7 the "squirrel" example which shows bird images. These should be fixed.
>
> Thank you for your effort! We will add the prompt descriptions in Figure 8 and will revise the wrong prompts (which should be "bird") in Figure 7. We will also carefully check typos and improve the clarity of the presented figures.

---

> > ### Comment · Reviewer_oEgS · 2025-08-05
> >
> > Thank you for the clarifications and additional experiments.
> >
> > Regarding complex prompts — I think the paper could benefit from more complex prompts comparisons with competing methods.
> >
> > Assuming the authors will indeed proofread the paper and fix typos and wrong captions, I will keep my rating.

---

> > > ### Author Response · Authors · 2025-08-05
> > >
> > > Thanks again for your time and effort in reviewing our paper. We will carefully proofread the paper and fix typos in our revision.

---

### Official Review · Reviewer_JABx · 2025-06-29

**Clarity:** 4
**Significance:** 3
**Originality:** 3
**Rating:** 5
**Confidence:** 3

**Summary:**

This paper introduces a finetuning method that enables few-shot, preference-aligned text-to-image generation using an off-the-shelf pretrained diffusion model. The authors critically examine the importance and difficulties of reward maximization in the setting of diffusion models and alignment to human preferences, and they develop a novel yet simple objective to enable alignment and few-shot sampling. Reward-Instruct shows strong results in effectively maximizing human preference, per a set of image-based reward models, however the generalization capabilities of the final model, i.e. alignment w.r.t to reward models not seen in training, remains unclear.

**Questions:**

- Could the authors please provide results that measure preference alignment using a reward model that was not used during Reward-Instruct’s training? I acknowledge the Aesthetics results but the Aesthetics model is biased towards simple image aesthetics and may not capture much of the intricacies of preference alignment in the context of text-to-image generation. I think using Pick-a-Pic reward model could be a good experiment.

- How does Reward-Instruct compare with Diffusion-DPO and/or KTO SD1.5 or other offline approaches?

- Could the authors please elaborate on the design decisions when selecting reward models? The justification for reward ensembling highlights that maximizing the reward based on shared data modes across different reward models can help avoid reward hacking. In the paper however, the reward models used are somewhat similar in that they are optimized for some combination of image aesthetics and/or image-text alignment. Does this mean the reward models must be “sufficiently similar” in terms of what they reward? As a hypothetical example, if you included a safety classifier as one of the reward models would the training still be stable and, most importantly, effective?

**Ethical Concerns:**

["NO or VERY MINOR ethics concerns only"]

**Final Justification:**

The paper presents an effective method that combines preference alignment and few-shot sampling. The results in the paper and additional results presented in the rebuttal validate the significance of these results. I believe the novelty, applicability, and effectiveness of this method merits acceptance.

**Limitations:**

Yes

**Quality:**

3

**Strengths And Weaknesses:**

**Strengths**

- Strong results w.r.t to few-step baselines and 50-step baselines

- Principled formulation shows how to enable reward maximization and even shows how to extend guidance into intermediate denoising steps. Also demonstrates the benefit of reward ensembling.

- Show extensions to image editing and super-resolution.

- Paper is well written and motivations are clearly expressed.


**Weaknesses**

- Generalization of the final model is unclear. The authors seem to maximize the combination of HPS, ImageReward, and CLIP scores (I would appreciate it if the authors could confirm this and also clarify which HPS reward model in particular). However, their evaluations measure the scores from these same reward models plus the LAION Aesthetics model. Thus, it’s unclear if the preference alignment generalizes outside of these reward models– e.g. how do the results compare if measuring Pick-a-Pic reward model score, or a different version of HPS rm, or ideally if using human annotators?

- Comparisons to reward-centric approaches. Many of the reported baselines, e.g. ReFL, optimize their model w.r.t to one reward signal whereas the benefit of ensembling rewards is a key finding of this paper. I understand that re-training such baselines may be infeasible, however I think it should be noted that the method benefits from multiple signals whereas many other baselines do not, when in fact their performance may improve with multiple signals too.

- Comparisons to offline approaches. Given the prevalence of offline approaches like Diffusion-DPO, how does Reward-Instruct compare to SD-1.5 aligned via Diffusion-DPO or Diffusion-KTO?

---

> ### Author Rebuttal · Authors · 2025-07-31
>
> We sincerely thank you for the valuable feedback. We address each comment below.
>
> > Generalization of the final model is unclear. The authors seem to maximize the combination of HPS, ImageReward, and CLIP scores (I would appreciate it if the authors could confirm this and also clarify which HPS reward model in particular). However, their evaluations measure the scores from these same reward models plus the LAION Aesthetics model. Thus, it’s unclear if the preference alignment generalizes outside of these reward models– e.g., how do the results compare if measuring Pick-a-Pic reward model score, or a different version of HPS rm, or ideally if using human annotators?
>
> Great suggestion. We maximize HPS V2.1, ImageReward, and Clip Scores. To fully confirm the effectiveness of our Reward-Instruct in generating high-quality samples, we conduct a user study between the base model (50 NFE) and Reward-Instruct (4 NFE). The user study is conducted by presenting users with two anonymous images generated by different models and asking them to select the sample with higher image quality and better prompt alignment. We randomly selected 20 prompts for image generation. In total, we collected approximately 20 user responses on 20 pairs.
>
> | Model      | Backbone | NFE  | HPS v2.0↑ | Pick Score↑ | User Preference↑ |
> | ---------- | -------- | ---- | --------- | ----------- | ---------------- |
> | Base Model | SD-v1.5  | 50   | 27.22     | 22.19       | 41.7%            |
> | RI (Ours)  | SD-v1.5  | 4    | 28.83     | 22.40       | 58.3%            |
>
> It can be seen that Reward-Instruct is able to generalize to unseen rewards, and also achieves better performance regarding user preference compared to the pre-trained base models.
>
> >Comparisons to reward-centric approaches. Many of the reported baselines, e.g. ReFL, optimize their model w.r.t to one reward signal whereas the benefit of ensembling rewards is a key finding of this paper. I understand that re-training such baselines may be infeasible, however I think it should be noted that the method benefits from multiple signals whereas many other baselines do not, when in fact their performance may improve with multiple signals too.
>
> Great suggestion. This is a very interesting direction. Indeed, some other RL methods that integrate multiple rewards could also potentially further improve performance. We will discuss this point in the revision and further explore it in future work. Nevertheless, it is worth noting that our reward-centric approach directly produces few-step samplers, without the need for diffusion distillation. In this sense, ReFL (standard diffusion sampling) is not a direct baseline in our setting.
>
>
>
> > Comparisons to offline approaches. Given the prevalence of offline approaches like Diffusion-DPO, how does Reward-Instruct compare to SD-1.5 aligned via Diffusion-DPO or Diffusion-KTO?
>
> We note that the DPO-like methods belong to another kind of method. We are also glad to provide the comparison between DPO and our Reward-Instruct below:
>
> | Model               | Backbone | NFE  | HPS↑  | Aes↑ | CS↑   | FID↓  | Image Reward↑ |
> | ------------------- | -------- | ---- | ----- | ---- | ----- | ----- | ------------- |
> | Base Model          | SD-v1.5  | 50   | 30.19 | 5.87 | 34.28 | 29.11 | 0.81          |
> | DPO                 | SD-v1.5  | 50   | 30.79 | 5.93 | 33.84 | 31.59 | 0.90          |
> | **Reward-centric:** |          |      |       |      |       |       |               |
> | ReFL                | SD-v1.5  | 50   | 31.82 | 5.97 | 31.78 | 39.38 | 1.16          |
> | RI (Ours)           | SD-v1.5  | 4    | 33.70 | 6.11 | 32.13 | 33.79 | 1.22          |
>
> We find that DPO typically aligns less with human preference compared to reward-centric approaches, but performs better regarding the FID. We believe that a possible reason is that DPO training still relies on curated data, which benefits the FID. However, it is clear that RI is not in conflict with DPO in that one can readily use a DPO-aligned diffusion model as a base model and apply RI after it to get the advantage of both. However, in this paper, we evaluate RI without the influence of DPO to clearly study its pros and cons.
>
> > Could the authors please elaborate on the design decisions when selecting reward models? The justification for reward ensembling highlights that maximizing the reward based on shared data modes across different reward models can help avoid reward hacking. In the paper however, the reward models used are somewhat similar in that they are optimized for some combination of image aesthetics and/or image-text alignment. Does this mean the reward models must be “sufficiently similar” in terms of what they reward? As a hypothetical example, if you included a safety classifier as one of the reward models would the training still be stable and, most importantly, effective?
>
> We believe that rewards do not need to be sufficiently similar to each other, and that the model may benefit from combinations of different types of rewards. In Table 3 of the Appendix, we maximize HPS, CLIP scores, and AeS, where each reward represents a different type. In particular,  HPS represents human preference, CLIP score represents image-text alignment, and AeS represents pure aesthetics. The results in Table 3 demonstrate that this combination can achieve similar rewards as well as notably better zero-shot FID, suggesting better avoidance of reward hacking. To further verify this, we took your valuable suggestion and incorporated a "safety classifier" into our training.
>
> | Model                               | Backbone | Steps | HPS↑  | Aes↑ | CS↑   | FID↓  | Image Reward↑ |
> | ----------------------------------- | -------- | ----- | ----- | ---- | ----- | ----- | ------------- |
> | RI w/ HPS + Clip Score + AeS (Ours) | SD-v1.5  | 4     | 32.35 | 6.21 | 32.43 | 31.26 | 1.01          |
> | RI, w/ safety checker               | SD-v1.5  | 4     | 33.62 | 6.08 | 32.47 | 33.51 | 1.18          |
> | RI (Ours)                           | SD-v1.5  | 4     | 33.70 | 6.11 | 32.13 | 33.79 | 1.22          |
>
> Results in above show that after integrating the safety classifier into the training, we can still achieve comparable performance and slightly better zero-shot FID.

---

> ### Comment · Reviewer_JABx · 2025-08-05
>
> Hi,
>
> I would like to thank the authors for their work and answers to my questions. However, regarding my first question about generalization, I still have some reservations about the provided results. First, wrt the automated metrics (HPS, etc.), HPS v2.1 is trained using the HPSv2.0 dataset so it would seem that maximizing HPSv2.1 would maximize HPSv2.0 score. However, the improvement in PickScore does seem promising as most papers usually show ~+0.2 improvement. My biggest issue is that the results do not provide enough information to gauge statistical significance which is important given that these are probabilistic models. The user study only compares 20 images which seems very small and it's hard to tell if the win-rate is significant. Also, this user study seems to have only gathered 1 response per image pair. I believe the questions about the generality of this method still remain.
>
> I would like to hear the authors' thoughts on these questions before I finalize my review. Thank you.

---

> > ### Author Response · Authors · 2025-08-05
> >
> > Thanks for the chance to further clarify our work.
> >
> > In order to assess the significance of the observed improvements, we performed paired t-tests on the results. The null hypothesis for these tests was that the performance of base model would not be inferior to that of Reward-Instruct. The outcomes of these statistical tests are shown below, where the plus symbol denotes levels of statistical significance for rejecting the null hypothesis, with +, ++, and +++ corresponding to significance levels of 5\%, 1\%, and 0.1\%, respectively.
> >
> > | HPS v2.0 | Pick Score | User Preference |
> > | -------- | ---------- | --------------- |
> > | +++      | ++         | +++             |
> >
> > The results indicate that Reward-Instruct achieves statistically significant improvement compared with the base model.
> >
> > We clarify that each user is asked to compare 20 image-prompt pairs rather than a single pair. This forms nearly 400 effective comparisons. We argue that because users are independent, the performance of shared versus different sets of 20 random prompts is likely to be similar.
> > We will conduct the user study on more prompts and report the results in the revision.

---

> > > ### Comment · Reviewer_JABx · 2025-08-05
> > >
> > > I thank the authors for the additional results. All of my questions have been answered.
> > >
> > > However I would like to note that the point made about the user study does not justify running such a setup for the user study. The reason user studies should assign multiple users per task (in this case, multiple users to review the same pair of images) is to reduce bias and better model the consensus, as such a study is trying to approximate human preference broadly. I hope the authors take this into account when writing the revision.

---

> > > > ### Author Response · Authors · 2025-08-06
> > > >
> > > > Thank you for your reply. We are glad that our responses have answered your questions. We will discuss the limitation of our user study in the revision and try our best to improve the user study following your valuable suggestions. Thank you again for your time and effort in reviewing our paper.

---

### Official Review · Reviewer_qewY · 2025-07-03

**Clarity:** 3
**Significance:** 3
**Originality:** 4
**Rating:** 5
**Confidence:** 3

**Summary:**

This paper introduces Reward-Instruct, a novel framework for creating high-quality, few-step image generators w/o complex diffusion distillation losses. The core idea is to reframe the task as regularized reward maximization, where a few-step generator's parameters are directly optimized to satisfy reward functions. To prevent "reward hacking," the method employs simple regularization techniques utilizing the models base parameters to approximate the base image distribution.
The authors investigate how to balance different reward functions and their impact on the downstream model. The experimental evaluation demonstrates that this distillation-free approach achieves state-of-the-art performance, surpassing previous methods in visual quality and quantitative metrics

**Questions:**

- Could you elaborate further on the intuition behind $\hat{\epsilon}_\eta$ (L146)? Is this routed in prior literature on DM sampling?
- What is the importance of randomly sampling $\eta$ at different diffusion steps? Could you provide further analysis of the impact of that design choice?

**Ethical Concerns:**

["NO or VERY MINOR ethics concerns only"]

**Final Justification:**

Taking into consideration the rebuttal response and author reviewer feedback I retain my positive rating.

**Limitations:**

Despite the authors claim in the NeurIPS checklist the paper does not discuss the limitations of the proposed method, nor do the authors provide insights on its failure cases.

**Quality:**

3

**Strengths And Weaknesses:**

# Strength

- Unique and novel perspective on reward modeling and DM distillation makes for a strong methodological contribution
- Rigorous methodological section with strong motivation and good mathematical background
- Interesting insights on reward hacking, as well as  balancing different reward function and their influence
- Experimental demonstration of effectiveness on 2 different DM paradigms: SD 1.5 for U-Net backbone and noise-estimate score function + SD3.5 for DIT backbone w/ rectified flow
- Qualitative and empirical performance improvements look solid
- Additional improvements in computational requirements

## Weaknesses
There remain some minor weaknesses that could be improved

- Many of the provided insights are only underpinned by a handful of cherry-picked examples (e.g. Fig. 4, Fig. 6, Fig. 7, Fig. 8, Fig. 11). The paper would benefit from corroborating these findings with quantitative experiments
- Use of LAION Aesthetics classifier v1 instead of v2, despite the improved version being readily available
- Inferring improved convergence from Fig. 10 (RewardInstruct+) is a bit of stretch. No improvement for HPS for example
- The presentation can be improved. Especially the sudden introduction of RewardInstruct+ over more substantive experiments may confuse some readers
- The paper would immensely benefit from a human user study on actual improvements in quality, since automated metrics are known to be faulty (and as discussed in the paper there may be reward hacking)

-

---

> ### Author Rebuttal · Authors · 2025-07-31
>
> We sincerely thank you for the valuable feedback. We address each comment below.
>
> > Many of the provided insights are only underpinned by a handful of cherry-picked examples (e.g. Fig. 4, Fig. 6, Fig. 7, Fig. 8, Fig. 11). The paper would benefit from corroborating these findings with quantitative experiments
>
> We provide quantitative results about removing one technique at a time to show the effectiveness.
>
> | Model                        | Backbone | Steps | HPS↑  | Aes↑ | CS↑   | FID↓  | Image Reward↑ |
> | ---------------------------- | -------- | ----- | ----- | ---- | ----- | ----- | ------------- |
> | RI (Ours)                    | SD-v1.5  | 4     | 33.70 | 6.11 | 32.13 | 33.79 | 1.22          |
> | RI w/ single reward          | SD-v1.5  | 4     | 32.08 | 5.80 | 31.01 | 38.21 | 0.89          |
> | RI w/o random-η-sampling     | SD-v1.5  | 4     | 33.41 | 6.12 | 32.11 | 36.90 | 1.18          |
> | RI w/o weight regularization | SD-v1.5  | 4     | 34.26 | 6.12 | 32.45 | 39.25 | 1.27          |
>
> The results, presented in above, demonstrate the critical role of each component:
>
> 1) Multiple rewards: Improves both rewards and FID, mitigating artifacts and reward hacking.
> 2) Random-$\eta$-sampling: Maintains similar reward performance but significantly improves FID, aiding to find better mode with fewer artifacts.
> 3) Weight regularization: Trades slight reward gains for better FID, ensuring the generator stays within the image manifold.
>
> > Use of LAION Aesthetics classifier v1 instead of v2, despite the improved version being readily available
>
> We clarify that AeS v2 is used in our experiments.
>
> > Inferring improved convergence from Fig. 10 (Reward Instruct+) is a bit of stretch. No improvement for HPS for example
>
> Reward-instruct only provides supervision at the endpoint and requires backpropagation to flow through the entire generator (multiple denoising steps), while Reward-instruct+ provides signal supervision during the process, which intuitively enables faster convergence and higher training efficiency, as we only allow RI+'s gradients to backpropagate one step rather than through the entire generator. Reward-instruct+ can converge faster than Reward-instruct regarding most rewards. Although it doesn't show significant gains on HPS, RI+'s training is cheaper (each iteration requires only 65\% of Reward-instruct's time), which makes it still have clear advantages over Reward-instruct overall. Exploring how to make RI+ have more consistent gains would be an interesting future direction.
>
> > The presentation can be improved. Especially the sudden introduction of Reward Instruct+ over more substantive experiments may confuse some readers
>
> Thank you for your valuable comments. We will add more motivating analysis and illustrations for the introduction of Reward-Instruct+ in our revision.
>
> > The paper would immensely benefit from a human user study on actual improvements in quality, since automated metrics are known to be faulty (and as discussed in the paper, there may be reward hacking)
>
> Great suggestion. We provide a user study to compare the base model and our reward-instruct. The user study is conducted by presenting users with two anonymous images generated by different models and asking them to select the sample with higher image quality and better prompt alignment. We randomly selected 20 prompts for image generation. Due to the limited time in the rebuttal phase, we collected approximately 20 user responses on 20 pairs in total.
>
> | Model      | Backbone | NFE  | User Preference↑ |
> | ---------- | -------- | ---- | ---------------- |
> | Base Model | SD-v1.5  | 50   | 41.7%            |
> | RI (Ours)  | SD-v1.5  | 4    | 58.3%            |
>
> Results in above shows that the proposed RI achieves better user preference compared to the base models, indicating the effectiveness of the proposed reward-centric approach in aligning human preference.
>
> > Could you elaborate further on the intuition $\epsilon_\eta$ behind (L146)? Is this routed in prior literature on DM sampling?
>
> This refers to an interpolation between the Consistency Model (CM) sampler and the DDIM sampler. The former is a purely stochastic sampler ($\eta=0$), while the latter is a deterministic sampler ($\eta$ = 1). There is a notable difference in both the sampling paths and the style of the resulting samples generated by these two distinct methods. Our experiments in Figure 4 further show that different $\eta$ leads to different styles.
> This discrepancy drives the strategy of interpolating between them by randomizing the eta parameter during sampling.
>
> > What is the importance of randomly sampling $\eta$ at different diffusion steps? Could you provide further analysis of the impact of that design choice?
>
> Randomly sampling $\eta$ at different diffusion steps can augment the generator's output distribution and broaden the scope of its exploration, since different $\eta$ deliver different sampling trajectories and styles. Consequently, we believe this can force the generator to learn a more robust policy by maximizing the rewards across these varied sampling paths, which ultimately improves its robustness.
>
> If we fixed $\eta$ in sampling during training, the generator only needs to learn to maximize the reward under a specific sampling trajectory and style, which makes it easier for the generator to hack the reward.
>
> | Model         | Backbone | Steps | HPS↑  | Aes↑ | CS↑   | FID↓  | Image Reward↑ |
> | ------------- | -------- | ----- | ----- | ---- | ----- | ----- | ------------- |
> | RI (Ours)     | SD-v1.5  | 4     | 33.70 | 6.11 | 32.13 | 33.79 | 1.22          |
> | RI w/ $\eta$ = 0   | SD-v1.5  | 4     | 33.29 | 6.05 | 31.50 | 37.57 | 1.09          |
> | RI w/ $\eta$ = 0.7 | SD-v1.5  | 4     | 33.74 | 6.19 | 32.30 | 36.09 | 1.10          |
> | RI w/ $\eta$ = 1   | SD-v1.5  | 4     | 33.41 | 6.12 | 32.11 | 36.90 | 1.18          |
>
> The results above show that fixed $\eta$ leads to inferior results in both FID and reward metrics.

---

> > ### Comment · Reviewer_qewY · 2025-08-05
> > **Response**
> >
> > I'd like to thank the authors for their detailed response and urge them to adjust the paper based on the rebuttal discussion.
> >
> > The additional ablations, and user study should meaningfully improve the paper.
> >
> > >RI+'s training is cheaper (each iteration requires only 65% of Reward-instruct's time),
> >
> > This benefit is currently not presented very clearly. Adding this efficiency/convergence improvement would help to provide a clear motivation for RI+.

---

> > > ### Author Response · Authors · 2025-08-06
> > >
> > > Thank you for your reply. We will include the rebuttal discussion in our revision.
> > >
> > > Thank you for the appreciation of our RI+. We will add the discussion on efficiency/convergence improvement to highlight the benefit of our RI+. Thank you again for your time and effort in reviewing our paper.

---

### Official Review · Reviewer_Bbgh · 2025-07-03

**Clarity:** 3
**Significance:** 2
**Originality:** 2
**Rating:** 4
**Confidence:** 4

**Summary:**

This paper proposes Reward-Instruct, a reward-centric training approach designed to efficiently convert pre-trained diffusion models into reward-aligned few-step generative models without relying on diffusion distillation losses. The key claim is that reward maximization alone, guided by a stochastic Langevin dynamics-inspired optimization, can effectively yield high-quality few-step generation. Extensive experiments demonstrate strong quantitative and qualitative performance on text-to-image benchmarks, showcasing superior visual quality compared to existing distillation-based methods.

**Questions:**

Since Reward-Instruct directly maximizes rewards over a few-step sampling process without explicit diffusion distillation, can the resulting outputs still be interpreted as valid diffusion score functions? Or should the model be viewed simply as a general epsilon-conditioned generator lacking traditional diffusion properties?

Without explicit diffusion distillation, how can RI theoretically or empirically guarantee meaningful preservation of the original pretrained diffusion model’s trajectories?

Why exactly does Reward-Instruct provide increased robustness against reward hacking compared to methods incorporating explicit diffusion distillation losses (e.g., DI++, RG-LCM)? Could similar robustness be achieved simply by applying RI-style reward maximization after completing standard diffusion distillation training?

How sensitive is RI to variations in reward scales or reward noise? Can RI reliably maintain stability and image quality if rewards become noisy or biased?

Could authors provide experiments using Stable Diffusion XL (SDXL), currently a prevalent baseline, to fully understand RI’s practical advantages and limitations compared to state-of-the-art few-step diffusion baselines?

**Ethical Concerns:**

["NO or VERY MINOR ethics concerns only"]

**Final Justification:**

During rebuttal period, the author actively engage in discussion, and address most of my concerns. Therefore, I vote positive score.

**Limitations:**

RI relies strictly on differentiable reward signals, potentially restricting applicability to black-box or human-derived feedback scenarios.

The method significantly deviates from traditional diffusion model inference procedures, compromising interpretability and the theoretical underpinnings of diffusion sampling.

RI’s robustness against reward hacking is claimed without rigorous theoretical or empirical validation, limiting the trustworthiness of this key assertion.

**Quality:**

3

**Strengths And Weaknesses:**

Strength

\* Reward-Centric Simplicity: The proposed approach elegantly bypasses the computational complexity and implementation difficulties inherent in diffusion distillation.

\* Strong Empirical Results: Impressive visual quality and improved performance metrics (FID, HPS, CLIP scores) across various text-to-image tasks.

\* Robustness via Multiple Rewards: Incorporates multiple reward signals effectively to mitigate common reward hacking pitfalls, achieving better image quality and stability.

Major Weakness

\*Conceptual Ambiguity Regarding Diffusion Sampling:
The proposed training method updates parameters by directly maximizing rewards in a few-step generation process, significantly deviating from standard diffusion score matching procedures. While the model structure still resembles diffusion-based epsilon-prediction, the learned outputs may no longer represent valid diffusion score functions. Thus, it is debatable whether the final model should be categorized as a diffusion model or rather a general generator shaped merely in the mathematical form of an epsilon predictor. The paper fails to clearly acknowledge or analyze this conceptual discrepancy.

\*Lack of Justification Regarding Diffusion Trajectory Preservation:
Traditional diffusion distillation methods explicitly attempt to preserve the trajectories defined by a pretrained diffusion model, enabling stable and theoretically grounded sampling dynamics. Reward-Instruct completely omits trajectory-level constraints, potentially collapsing diffusion trajectories into reward-driven but arbitrary generation paths. Without any formal analysis or justification, it remains unclear whether this approach maintains meaningful diffusion trajectories or merely exploits reward-driven shortcuts.

\*Insufficient Justification of Reward Hacking Robustness:
The paper claims RI’s robustness against reward hacking compared to diffusion-distillation-based methods like DI++ or RG-LCM. However, no clear theoretical explanation or thorough empirical demonstration is provided. Moreover, it is unclear why similar robustness could not be achieved simply by applying reward maximization after standard diffusion distillation.

\*Critical Omission of Relevant Recent Methods:
Several closely related recent works addressing reward-guided few-step diffusion methods are notably missing:
"[ICLR 2025] Tuning Timestep-Distilled Diffusion Model Using Pairwise Sample Optimization"
"[CVPR 2025] Reward Fine-Tuning Two-Step Diffusion Models via Learning Differentiable Latent-Space Surrogate Rewards"
"[ICLR 2025 Oral] DSPO: Direct Score Preference Optimization for Diffusion Model Alignment (DPO-based)"

Minor Weakness

\*Inconsistent Naming Conventions:
The method is inconsistently named (Reward-Instruct, RI, RO), creating confusion (e.g., Figure 5). Clearly choose and consistently use one abbreviation throughout the paper.

\*Unclear CFG Loss Description:
The algorithm presented in the appendix includes a term named "CFG Loss" without any explicit description or justification within the main body, making the motivation and necessity unclear.

\*Missing Algorithm Description for Reward-Instruct+:
Despite introducing Reward-Instruct+ (intermediate-step supervision), the paper lacks a clear, formal algorithmic presentation for this variant.

\*Lack of SDXL Comparison:
While experiments are conducted on Stable Diffusion v1.5 and SD3-Medium, no comparisons or experiments with SDXL—currently a prevalent and practically important baseline—are provided. Additionally, SD3-Medium results are presented without comparison to existing methods, reducing interpretability and value.

---

> ### Author Rebuttal · Authors · 2025-07-31
>
> We sincerely thank you for the valuable feedback. We address each comment below.
>
> > Since Reward-Instruct directly maximizes rewards over a few-step sampling process without explicit diffusion distillation, can the resulting outputs still be interpreted as valid diffusion score functions? Or should the model be viewed simply as a general epsilon-conditioned generator lacking traditional diffusion properties?
>
> The generator trained by Reward-Instruct is no longer diffusion score functions. We note that almost all distilled diffusion models are also no longer diffusion score functions, since their objectives are not for learning scores.
>
> > Without explicit diffusion distillation, how can RI theoretically or empirically guarantee meaningful preservation of the original pretrained diffusion model’s trajectories?
>
> Currently, there are usually two mainstream diffusion distillation approaches: one is distribution matching, which aims to align the student and teacher at the distribution level (such as SD-Turbo [48] and Diff-instruct [28]); the other is trajectory distillation, which aims to simulate the original diffusion model’s trajectories with fewer steps (such as Consistency Models [5] and CTM [21]). Only the latter aims to preserve the original diffusion model’s trajectories, while the former performs much better than the latter and is the most popular approach currently. Our method, different from traditional diffusion distillation approaches, is a novel reward-centric approach. Note that, similar to distribution matching, our approach achieves strong performance without preserving the original diffusion model’s trajectories.
>
> It is also worth emphasizing that there has recently been a paradigm shift from supervised training (data-centric) to RL training (reward-centric). In the LLM field, the community is actively exploring reward-centric post-training approaches that can demonstrate powerful emergent capabilities, e.g., the GRPO adopted in DeepSeek R1. **Our work represents a systematic study of the first reward-centric approaches to post-train pre-trained diffusion models into powerful few-step generators.** By our reward-centric RI, a simple yet elegant approach that doesn't rely on traditional diffusion distillation techniques, can achieve surprisingly good few-step generation performance.
> We hope our investigation is helpful for the community to make further explorations along this novel path.
>
> > Why exactly does Reward-Instruct provide increased robustness against reward hacking compared to methods incorporating explicit diffusion distillation losses (e.g., DI++, RG-LCM)? Could similar robustness be achieved simply by applying RI-style reward maximization after completing standard diffusion distillation training?
>
> Our work aims to explore a reward-centric approach to achieve fast generation without the need for tricky yet expensive diffusion distillation. To achieve this, we investigate several simply yet effective regularizations to alleviate reward hacking, which could also potentially benefit the distillation-based reward maximization for better performance.
> It is our experimental observation that our RI delivers surprisingly robust performance with simple regularization.
> However as you suggested, we agree that it is exciting to further explore the theoretical properties of regularizations introduced with RI in the future.
>
> Interestingly, we can explore combining reward-centric RI training with traditional distillation in our future work to produce more powerful models.
>
> Starting from the distilled model is a much stronger initialization compared to pre-trained diffusion models. We can directly initialize from the distilled model, and replace the prior with the weight of the distilled model to construct RI for fine-tuning the distilled model. We show results below:
>
> | Model                  | Backbone | NFE  | HPS↑  | Aes↑ | CS↑   | FID↓  | Image Reward↑ |
> | ---------------------- | -------- | ---- | ----- | ---- | ----- | ----- | ------------- |
> | Base Model             | SD-v1.5  | 50   | 30.19 | 5.87 | 34.28 | 29.11 | 0.81          |
> | RI, init w/ LCM (Ours) | SD-v1.5  | 4    | 33.62 | 6.15 | 33.58 | 32.05 | 1.19          |
> | RI (Ours)              | SD-v1.5  | 4    | 33.70 | 6.11 | 32.13 | 33.79 | 1.22          |
>
> It can be seen that RI can effectively fine-tune the distilled models with better performance compared to initializing from pre-trained models.
>
> > Several closely related recent works addressing reward-guided few-step diffusion methods are missing: PSO [a], LaSPO [b], DSPO [c].
>
> Thank you for bringing our attention to these interesting works. We discuss the mentioned works below:
>
> - **DSPO, PSO, and LaSPO focus on essentially different tasks compared to us**. DSPO focus on fine-tuning pre-trained diffusion models, which is not for constructing few-step models. PSO and LaSRO aim to fine-tune the well-trained few-step models, which is a different and much easier setting compared to our method and our baselines. **The notable contribution and finding of our work is that: we show that reward-centric approach can fine-tuning form pre-trained diffusion models to few-step generator.**
>
> - **PSO and DSPO belong to DPO-like methods, which are essentially different from RL methods** that aim to maximize a certain reward. In their papers [a, c], there is no comparison between DPO-like methods and RL-based methods.
>
> - However, we are also glad to provide a comparison below:
>
> | Model                  | Backbone | NFE  | HPS↑  | Aes↑ | CS↑   | FID↓  | Image Reward↑ |
> | ---------------------- | -------- | ---- | ----- | ---- | ----- | ----- | ------------- |
> | Base Model             | SD-v1.5  | 50   | 30.19 | 5.87 | 34.28 | 29.11 | 0.81    |
> | DSPO                   | SD-v1.5  | 50   | 30.76 | 5.96 | 33.81 | 30.86 | 0.92   |
> | PSO, init w/ LCM       | SD-v1.5  | 4    | 29.24 | 5.99 | 32.03 | 33.24 | 0.85   |
> | LaSPO, init w/ LCM     | SD-v1.5  | 4    | 33.16 | 6.13 | 30.83 | 34.73 | 0.94   |
> | RI, init w/ LCM (Ours) | SD-v1.5  | 4    | 33.62 | 6.15 | 33.58 | 32.05 | 1.19|
> | RI (Ours)              | SD-v1.5  | 4    | 33.70 | 6.11 | 32.13 | 33.79 | 1.22|
>
> It can be seen that applying reward-instruct for post-training distill model (LCM) is more effective than starting from pre-trained diffusion models, while reward-instruct init w/ LCM outperforms the prior works in fine-tuning LCM regarding both reward metrics and FID.
>
> [a] Tuning Timestep-Distilled Diffusion Model Using Pairwise Sample Optimization
>
> [b] Reward Fine-Tuning Two-Step Diffusion Models via Learning Differentiable Latent-Space Surrogate Rewards
>
> [c] DSPO: Direct Score Preference Optimization for Diffusion Model Alignment
>
> > How sensitive is RI to variations in reward scales or reward noise? Can RI reliably maintain stability and image quality if rewards become noisy or biased?
>
> For reward scales, we have applied gradient normalization to balance different rewards, which effectively balances contributions from different rewards regardless of their individual scales. Hence, our method is robust to reward scales. We have verified the effectiveness of gradient normalization in Figure 9.
>
> In the case of biased rewards, we expect RI with multiple rewards to be also relatively robust.
> On one hand, if only one reward is biased, its negative effect will not be significant since the supervision is an average of all rewards.
> On the other hand, even if all rewards are biased, we believe that as long as the biases are not consistent (e.g., biases are random with zero mean), their average effect should be neutral.
>
> It would be an interesting future work to conduct exact experiments about this matter. However, such an investigation is difficult to conduct in a rigorous manner since, in the current community, it is difficult to obtain a large pool of rewards with different sources of quantifiable biases.
>
> > Inconsistent Naming Conventions
>
> RI is short for Reward-Instruct, and R0 is a typo. Sorry for the confusion.
> We will clean up these inconsistent abbreviations and use RI as our abbreviation.
>
> > Unclear CFG Loss Description
>
> It has been pointed out in [27] that CFG can be viewed as an implicit reward function. Using it in our RI incurs a slight difference compared with other rewards, since it does not have an explicit value.
>
> However, we can compute its gradient. Hence, the CFG loss is a surrogate loss that has the same gradient of $-E_{x_t,t}\log p_\psi(\mathrm{text}|x_t) = -E_{x_t,t}\log \frac{p_\psi(x_t|\mathrm{text})}{p_\psi(x_t|\emptyset)}$. We will add a detailed description in our revision.
>
> > Missing Algorithm Description for Reward-Instruct+
>
> Thank you for your suggestion.
> We have detailed our RI+ in Section 3.4, but didn't summarize it into an algorithm.
> We will add an algorithm description for Reward-Insturct+ in revision.
>
> > SD3-Medium results are presented without comparison to existing methods, reducing interpretability and value.
>
> We note that the multi-step SD3-Medium itself is a strong baseline.
> The comparison between multi-step SD3-Medium and 4-step RI shows that **RI, as a reward-centric approach, is capable of post-training powerful flow-based models to few-step generators with better reward metrics and comparable zero-shot FID**. This highlights RI's effectiveness and its wide application.
>
> > Lack of SDXL Comparison.
>
> We compare to the base model (SDXL) and distillation-based RL methods, i.e., RG-LCM and DI++.
>
> | Model | NFE | HPS↑ | Aes↑ | CS↑ | FID↓ | Image Reward↑ |
> |---------|-----|--------|--------|--------|---------|----------------|
> | Base | 50 | 33.62 | 6.24 | 36.15 | 28.61 | 1.12 |
> | RG-LCM | 4 | 32.5 | 6.08 | 32.96 | 36.13 | 1.27 |
> | DI++ | 4 | 33.96 | 6.13 | 32.75 | 34.02 | 1.35 |
> | RI (Ours) | 4 | 34.83 | 6.30 | 34.50 | 32.45 | 1.40 |
>
> The results shown above show that the proposed Reward-centric approach - Reward-instruct also has advantages over prior distillation-based RL methods on the SDXL backbone.

---

> > ### Comment · Reviewer_Bbgh · 2025-08-04
> >
> > I appreciate the authors' efforts in addressing most of my concerns. However, several key issues remain unresolved
> > - There exists a discrepancy between the experimental setups - the author evaluates PSO and LaSRO on Stable Diffusion 1.5, whereas the original PSO/LaSRO papers report results on SDXL. This inconsistency undermines the validity of comparative analyses. Furthermore, since LaSRO's implementation is not publicly available, the reproducibility of these results is questionable.
> > - I remain unclear about the mechanism behind the claimed robustness against reward hacking. Could you please clarify:
> > Is the robustness primarily attributable to the SGLD regularization? If so, what specific properties of SGLD contribute to this effect?
> > - How exactly is the SGLD update incorporated within the backward pass? Please provide pseudo-code showing the complete optimization step, including:
> > ```python
> > loss = reward_function(model, target)
> > loss.backward()
> > # Detailed SGLD implementation here
> > ```

---

> > > ### Author Response · Authors · 2025-08-05
> > > **Further response (2/2)**
> > >
> > > > I remain unclear about the mechanism behind the claimed robustness against reward hacking. Could you please clarify: Is the robustness primarily attributable to the SGLD regularization?
> > >
> > > It is worth emphasizing that the robustness of our RI is mainly demonstrated empirically.
> > > As one of our key contributions, we found from experiments that our RI is surprisingly robust to different choices of regularization -- various regularization techniques work surprisingly well.
> > > In the current stage, we do not have rigorous robustness analysis that makes it provable, mainly due to the fact that rewards themselves are black-boxes.  While outside the scope of this work, we will make further investigations to fully elucidate the mechanism in our future work. Nevertheless, we do have our understanding of RI's robustness and they served as our motivation when designing RI.
> > >
> > >
> > >
> > > We believe the robustness is attributable to the combination of several proposed effective regulations in "weakening" the generator's capability, e.g., SGLD, random-$\eta$ sampling, and weight regularization. We have ablated the effectiveness of each component in Figures 6 (main paper), 8 (main paper), 13 (appendix), and 14 (appendix). We also report the ablated results by removing the techniques one by one over single reward (HPS v2.1) case below:
> > >
> > > | Model               | Backbone | Steps | HPS↑  | Aes↑ | CS↑   | FID↓   | Image Reward↑ |
> > > | ------------------- | -------- | ----- | ----- | ---- | ----- | ------ | ------------- |
> > > | RI w/ single reward | SD-v1.5  | 4     | 32.08 | 5.80 | 31.01 | 38.21  | 0.89          |
> > > | - random-η sampling | SD-v1.5  | 4     | 32.73 | 5.75 | 30.77 | 42.18  | 0.80          |
> > > | - Weight Prior      | SD-v1.5  | 4     | 33.59 | 5.73 | 30.45 | 61.49  | 0.71          |
> > > | - SGLD              | SD-v1.5  | 4     | 36.08 | 5.41 | 25.80 | 107.63 | 0.45          |
> > >
> > > It can be seen that each regularization technique leads to worse HPS, but notably better other rewards and FID. This highlights the effectiveness of the proposed regularization in over-optimizing rewards for preventing artifacts.
> > >
> > > Straightforwardly, SGLD aims to sample from the reward-tilted distribution, which we believe does not exhibit severe reward-hacking. The reward-titled distribution assigns a reasonable prior to weights, therefore effectively constrains the distribution of generation in the image domain. Besides, the Langevin dynamics introduces Gaussian noises to the optimization process that promotes exploration.  Other techniques like random-$\eta$-sampling also contribute empirically by inspiring the generator to explore a broader area.
> > > We believe our regularizations explicitly and effectively inspire the generator with more diverse exploration to avoid over-optimizing the rewards, while distillation-based approaches such as DI++ do not encourage exploring broader areas and assign weight prior.
> > >
> > > > How exactly is the SGLD update incorporated within the backward pass? Please provide pseudo-code showing the complete optimization step, including:
> > >
> > > We provide the torch-style pseudo code for implementing the SGLD and weight regularization below:
> > >
> > > ```
> > > loss.backward()
> > > for param, param_0 in zip(model.parameters(), model_pretrained.parameters()):
> > >     with torch.no_grad():
> > >         if param.grad is not None:
> > >             # SGLD noise:
> > >             noise_para = torch.randn_like(param.grad.data) * noise_scale
> > >             # weight regularization:
> > >             reg_para = (param - param_0) * reg_scale
> > >             param.grad.data.add_(noise_para)
> > >             param.grad.data.add_(reg_para)
> > > optimizer.step()
> > > ```

---

> > > > ### Comment · Reviewer_Bbgh · 2025-08-06
> > > >
> > > > Thanks for your response. Your reply addresses most of my concerns, but I still think the manuscript could be clearer for readers. In particular, I have one implementation‑detail question about the pseudo‑code:
> > > >
> > > > How is reg_scale calculated? In the paper we have
> > > > reg_scale = learning_rate / (2 * sigma_t).
> > > > How is sigma_t determined? Is it sampled from the final diffusion step among the timesteps?
> > > > For example, with a 4‑step scheduler where the timesteps are {999, 819, 619, 419}, does sigma_t come from the initial index (999) or the final index (419)?

---

> > > > > ### Author Response · Authors · 2025-08-06
> > > > >
> > > > > Thanks for your reply. We are glad that our response has addressed most of your concerns.
> > > > >
> > > > > To further clarify, the reg_scale of weight prior is calculated by $\frac{1}{2\sigma^2}$ without learning rate, where $\sigma$ is a fixed hyper-parameter. In practice, the learning rate would be handled inside the optimizer's step. As a general rule of thumb, we found that letting $\frac{1}{2\sigma^2}$ be $0.1$ or $1$ is effective in practice.
> > > > >
> > > > > We note that the $\sigma$ is not related to the timestep $t$. Hence, it is shared among all steps. Specifically, the weight prior directly operates the models' weight, which is not related to the forward/reverse diffusion process.
> > > > >
> > > > > Hope this could make things clearer. Thank you again for your time and effort in reviewing our paper.

---

> > > > > > ### Comment · Reviewer_Bbgh · 2025-08-06
> > > > > >
> > > > > > Thanks for the clarification. I now understand the role of the reg_scale, and I appreciate the authors for the clear explanation.
> > > > > >
> > > > > > While I see that reg_scale is typically set to 0.1 or 1, it seems that the noise scale for SGLD is not explicitly mentioned. Could the authors clarify how the noise scale is determined or computed in practice? It would be helpful to include this in the final manuscript.
> > > > > >
> > > > > > If the authors could clearly specify this point, I would be happy to consider raising my score.

---

> > > > > > > ### Author Response · Authors · 2025-08-06
> > > > > > >
> > > > > > > Thanks for your timely reply. We would like to further clarify that the noise scale in our pseudo code is computed by $\frac{\sqrt{2\mathrm{\lambda}}}{\lambda}$, where $\lambda$ denotes the learning rate. The reason for dividing by $\lambda$ is that it would be multiplied by the learning rate $\lambda$ inside the optimizer's step, which ultimately results in the added noise being $\sqrt{2\lambda}\epsilon$ ($\epsilon\sim N(0,I)$) as specified by the Langevin dynamics.
> > > > > > >
> > > > > > > Hope this could make things clearer. We will include this in our revision to make it clearer. Thanks again for your time and effort in reviewing our paper.

---

> > > > > > > > ### Comment · Reviewer_Bbgh · 2025-08-07
> > > > > > > >
> > > > > > > > Thanks for kind response. Please include that in your revision. I happy to raise my score!.

---

> > > > > > > > > ### Author Response · Authors · 2025-08-07
> > > > > > > > >
> > > > > > > > > Thanks for your timely reply and for raising score! We will include the discussion in our revision following your valuable suggestions. Thanks again for your time and effort in reviewing our paper.

---

> > > ### Author Response · Authors · 2025-08-05
> > > **Further response (1/2)**
> > >
> > > Thank you for the chance to further justify our work. We provide further response below.
> > >
> > > > There exists a discrepancy between the experimental setups - the author evaluates PSO and LaSRO on Stable Diffusion 1.5, whereas the original PSO/LaSRO papers report results on SDXL. This inconsistency undermines the validity of comparative analyses. Furthermore, since LaSRO's implementation is not publicly available, the reproducibility of these results is questionable.
> > >
> > > - We highlight again that our ***Reward-Instruct is the first reward-centric approach that can fine-tune the pre-trained diffusion models to few-step generators***. We believe this denotes a valuable finding and a significant contribution to the NeurIPS community.
> > > - Nevertheless, we provide further comparison on SDXL by directly taking their reported results from the original paper.
> > > - Compared to PSO. We note that our training does not include the pick score, while online-PSO constructs pairwise samples by the pick score for optimization.
> > >
> > > | **Dataset**       | **Method**               | **NFE**↓ | **Pick Score**↑ | **CLIP Score**↑ | **Image Reward**↑ | **AeS**↑ |
> > > | ----------------- | ------------------------ | -------- | --------------- | --------------- | ----------------- | -------- |
> > > | **Pickapic Test** | SDXL                     | 50       | 22.30           | 37.13           | 0.86              | 6.06     |
> > > |                   | DMD2                     | 4        | 22.35           | 36.79           | 0.94              | 5.94     |
> > > |                   | Online-PSO, init w/ DMD2 | 4        | 22.73           | 36.71           | 0.98              | 6.08     |
> > > |                   | Ours, init w/ DMD2       | 4        | 22.91           | 37.08           | 1.25              | 6.25     |
> > > | **Parti-Prompts** | SDXL                     | 50       | 22.77           | 36.07           | 0.91              | 5.75     |
> > > |                   | DMD2                     | 4        | 22.99           | 36.07           | 1.07              | 5.67     |
> > > |                   | Online-PSO, init w/ DMD2 | 4        | 23.29           | 36.34           | 1.17              | 5.84     |
> > > |                   | Ours, init w/ DMD2       | 4        | 23.80           | 36.56           | 1.33              | 6.07     |
> > >
> > > It can be seen that our RI achieves better results than PSO on all metrics, including the pick score unseen to RI but used to construct pairwise samples for the training of PSO.
> > >
> > > - Compared to LaSRO. It should be noted that we use HPS and clip score for training our RI in the following table, except for the Image Reward for fair comparison.
> > >
> > > | **Method**                | **NFE** | **Image Reward**↑ |
> > > | ------------------------- | ------- | ----------------- |
> > > | SDXL-Turbo                | 2       | 0.84              |
> > > | LaSRO, init w/ SDXL-Turbo | 2       | 0.96              |
> > > | Ours, init w/ SDXL-Turbo  | 2       | 1.13              |
> > >
> > > It can be seen that our RI notably outperforms LaSRO regarding the Image Reward that is unseen to both methods.

---

### Note · Authors · 2025-08-13

We thank the area chair and all reviewers for your time, insightful suggestions, and valuable comments. Your suggestions have been invaluable in refining our work, and we deeply appreciate the time and effort you dedicated to reviewing our paper. We have carefully addressed all points in our response.

We are encouraged by the reviewers’ positive feedback on various aspects of our work:

- Reviewer Bbgh:  Reward-Centric Simplicity: The proposed approach elegantly bypasses the computational complexity and implementation difficulties inherent in diffusion distillation.

- Reviewer qewY: Unique and novel perspective on reward modeling and DM distillation makes for a strong methodological contribution

- Reviewer JABx: Principled formulation shows how to enable reward maximization and even shows how to extend guidance into intermediate denoising steps. Also demonstrates the benefit of reward ensembling.

- Reviewer oEgS: The authors offer an interesting re-formulation of image generation formula post pre-training by placing the reward in the center.

 We are glad to have resolved concerns from reviewers, with our clarifications and rebuttal materials being well-received. We also appreciate the reviewers' constructive comments, which have allowed us to improve the clarity and comprehensiveness of our work.

In summary, our work represents ***a systematic study of the first reward-centric approaches to post-train pre-trained diffusion models into powerful few-step generators***. We believe this denotes a valuable finding and contribution to the NeurIPS community.

Thank you again for your insightful feedback and for the effort in reviewing our paper.

---

### Decision · Program_Chairs · 2025-09-17

**Decision:**

Accept (poster)

**Comment:**

The reviewers engaged in extensive discussion with the authors. Following the rebuttal, all reviewers reached a consensus in favor of acceptance. The AC finds no major outstanding concerns, aside from improving the writing and clarifying certain points in revision. Therefore, the AC agrees with the reviewers and recommends acceptance.